# Ocean acidification at the Toarcian Anoxic Event captured by boron isotopes in the lime mud record

Simone A. Kasemann [1] ✉, Tina Klein[1], Richard A. Boyle [2], Clemens V. Ullmann [3], Martin Aberhan[4], Anette Meixner[1], Luís V. Duarte [5], Timothy M. Lenton [2], Veronica Piazza[4] & Rachel A. Wood [6]

The Toarcian Oceanic Anoxic Event (ca. 183 million years ago) marks a global mass extinction coincident with dramatic changes in climate and ocean circulation, likely driven by large igneous province emplacement. Rapid carbon dioxide release may have induced global warming, widespread ocean deoxygenation, and ocean acidification. To constrain the magnitude of ocean acidification, we present boron isotope data from three different carbonate components, lime mud (micrite), brachiopods, and bivalves, from two marine sections in SW Europe. Only data from micrite shows a temporary decrease in boron isotope composition during the Toarcian Oceanic Anoxic Event, recording an ocean acidification event, which we reproduce using a coupled biogeochemical model. The contrasting stability of boron isotope values shown by bivalves and brachiopods suggests that the investigated taxa may have been able to physiologically buffer changes in ocean pH, and are therefore poor targets for the interrogation of pH changes in Earth history.

The Early Jurassic Toarcian Oceanic Anoxic Event (T-OAE) comprised an episode characterised by widespread dysoxic to anoxic/euxinic marine conditions[1], accompanied by dramatic changes in both pelagic and benthic communities e.g.[2–5]. The environmental changes and biological crisis are linked to pulses of rapid intrusive and extrusive volcanism and contact metamorphism associated with the Karoo and, in particular, the Ferrar large igneous province (LIP)[6–9]. This led to a rapid elevation of atmospheric carbon dioxide ($CO_2$) concentrations[10,11], from ca. 260 ppm[12] to as high as 1000 ppm[11] or even 1800 ppm[10], then a return to pre-event values[13,14]. One consequence of the rise in volcanogenic $CO_2$ together with the potential release of methane ($CH_4$) from this and other sources[10,15–17] was rapid greenhouse warming, with modelled increases in atmospheric temperatures of ca. 5 °C at mid-latitudes[13], and from ca. 3 to 7 °C for seawater[14]. These effects are documented by a globally recognized large-magnitude negative carbon isotope excursion (CIE, $\delta^{13}C$) observed in fossil wood, diverse marine bulk organic and inorganic substrates, and carbonate macrofossils e.g.[4,14,16,18,19]. Consistent with the global expression of the negative CIE and following these studies, the T-OAE is defined here chemostratigraphically to range from the onset of the negative CIE to the end of the recovery of $\delta^{13}C$ values (but see Erba et al.[20] for an alternative definition). This interval has been recently calibrated to last about 500,000 years[21]

consistent with the timescale for $CO_2$ removal by enhanced silicate weathering[22].

An associated effect of a rapid input of $CO_2$ in the atmosphere–ocean system is ocean acidification[23]. Evidence for ocean acidification has been inferred from a carbonate production crisis, as shown by a decrease in the carbonate accumulation rate[24,25], and probable decrease in seawater carbonate saturation[19,26,27] combined with a documented decrease in nanno-plankton flux[26]. Boron isotope data ($\delta^{11}B$) obtained from brachiopods from the Lusitanian Basin in Portugal have also been interpreted to indicate a decrease in seawater pH for the T-OAE interval[28]. In addition, the T-OAE extinction was notably selective against physiologically-unbuffered organisms, i.e., calcifiers and hypercalcifiers such as corals and bivalves, arguing for ocean acidification as a direct cause of extinction[25,29].

A size reduction in pelagic and benthic marine calcifying organisms and assemblages has been documented in association with the T-OAE[5,19,30] and attributed to various environmental factors, broadly coincident with the inferred ocean acidification. However, an apparent temporal offset between the observed changes in calcium carbonate content, nannofossil flux and size, and the decrease in $\delta^{11}B_{brachiopod}$-ocean pH has been used to argue against ocean acidification being responsible for the demise of the pelagic carbonate factory[31]. In addition, stable Ca and Sr isotope records from

[1]Faculty of Geosciences and MARUM—Center for Marine Environmental Sciences, University of Bremen, Bremen, Germany. [2]Global Systems Institute, University of Exeter, Exeter, UK. [3]Camborne School of Mines & Environment and Sustainability Institute, University of Exeter, Penryn, UK. [4]Museum für Naturkunde, Leibniz Institute for Evolution and Biodiversity Science, Berlin, Germany. [5]University of Coimbra & MARE, Coimbra, Portugal. [6]School of GeoSciences, University of Edinburgh, Edinburgh, UK. ✉e-mail: kasemann@uni-bremen.de

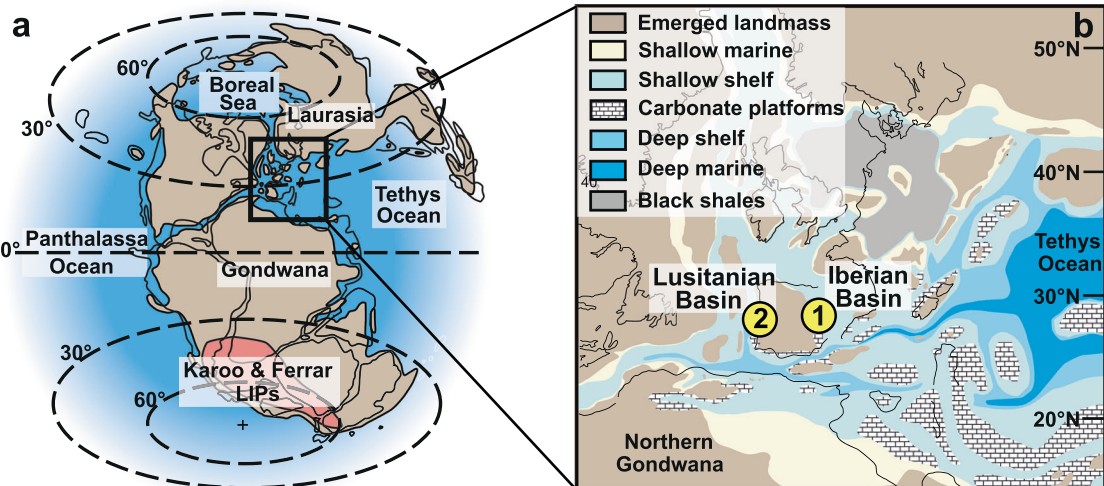

**Fig. 1 | Paleogeography of the Pliensbachian-Toarcian successions of the Iberian Peninsula. a** Paleogeographic reconstruction for the Early Jurassic (~180 Ma). **b** Locations of the two sampled sections in the surroundings of the Iberian Massif, Barranco de la Cañada, Spain (1) and Rabaçal/Fonte Coberta Portugal (2). Figures after Dera et al.[77]. and Thierry et al.[78].

belemnite and brachiopod skeletal carbonate (Yorkshire, UK, and Peniche, Portugal) through the early Toarcian do not appear to support ocean acidification[32]. These data highlight the difficulty of unambiguously attributing isotope and biogeochemical changes to changes in the pH of the ocean, exacerbated by the complex nature of multiple environmental perturbations associated with the T-OAE.

Quantification of the pH of ancient oceans from boron isotope composition of marine carbonates is always challenging due to the poorly constrained primary B isotope composition of ambient seawater ($\delta^{11}B_{SW}$), the mostly biologically driven isotope fractionation factors between the archive and seawater, changing environmental conditions, and the potential influence of diagenetic overprints (see Methods and Supplementary Discussion). Well-preserved rhynchonelliform brachiopods are often favoured for many geochemical analyses due to their dense, low-Mg calcite composition[14] and have been used for $\delta^{11}B$-ocean pH calculations for the Permo-Triassic[33] and Early Jurassic[28]. However, it remains unclear as to how the $\delta^{11}B$ values in such material in fact represent changes in ocean pH. While the calcite of some modern terebratulid brachiopod shells can be demonstrated to record changes in $\delta^{11}B$ values under changing ocean pH conditions, other brachiopod groups, however, biologically buffer the pH of their internal calcifying fluids, so leading to considerable vital effects that are species-specific[33–35] and which defy $\delta^{11}B$-ocean pH relationships and calibrations. Modern bivalves have a highly variable response to acidification, with different species exhibiting negative, neutral or even positive effects[36], and some having the ability to control the pH of their internal calcifying fluids[37–40]. Consequently, no $\delta^{11}B$-ocean pH calibration exists for bivalves.

By contrast, ancient lime mud, known as micrite, where grain sizes are up to 4 μm, have been demonstrated to offer a reliable archive in deep time for measuring B isotope values to track trends in ocean pH conditions when appropriately screened for contamination or diagenetic overprinting[41–44]. For the Pliensbachian and Toarcian, we speculate that algae, microbes, or most likely calcareous nannoplankton are the major sources of micrite; planktonic foraminifera had yet to evolve. Calcareous nannoplankton such as *Schizosphaerella*, coccoliths[19,26,45–47], and dinoflagellates[48–50] were abundant during this interval and may have produced most of the sampled micrite. The micrite might also represent an admixture of peri-platform ooze from a proximal carbonate platform, potentially supplying aragonitic fines. While relationships between the $\delta^{11}B$ value of calcite and seawater pH have been observed for some of the assumed source materials in the micrite (Supplementary Discussion), there is no useful $\delta^{11}B$-ocean pH calibration for this mixture of sample types, and we can therefore only track trends in ocean pH conditions.

Here we test the hypothesis of ocean acidification during the T-OAE interval by creating a multi-isotopic (boron, $\delta^{11}B$, carbon, $\delta^{13}C$, and oxygen, $\delta^{18}O$) record (Methods). Recognizing that the use of micrites and fossils as an archive of ocean pH conditions is a major challenge, we measure the values of different carbonate components, in order to derive the apparent pH change recorded in different target material. We compare the boron isotope record from three components of the carbonate rock record—micrite, well-preserved calcitic rhynchonellid brachiopods (including *Soaresirhynchia bouchardi*) and the dominantly calcitic bivalve (oyster) *Gryphaea*. This allows comparison of the response of two groups of calcifying marine benthos together with micrite across a time interval for which decreasing ocean pH is a plausible scenario. We follow strict diagenetic and contamination screening protocols for all components (Methods and Supplementary Discussion).

All samples (micrite and shells) were collected from two shallow marine carbonate successions deposited in the NW Tethys: Barranco de la Cañada in Spain, deposited in the Iberian Basin, and Rabaçal/Fonte Coberta in Portugal, deposited in the Lusitanian Basin (Fig. 1, Method and Supplementary Note 1). These show no evidence of widespread recrystallisation, deep-burial overprint, dolomitization, or meteoric diagenetic influence (Method). Sampling ranged from the latest Pliensbachian Margaritatus Zone to the mid-Toarcian Bifrons Zone. We integrate biostratigraphy (Supplementary Fig. 1 and Note 2) with $\delta^{13}C$ isotope chemostratigraphy to construct a common age model. Instead of solely using the B isotope records to reconstruct a proxy dataset of ocean pH as is commonly undertaken, we use a biogeochemical model of the coupled carbon, oxygen, phosphorus and sulfur cycles and evolution (COPSE) and their connecting feedbacks[51,52] (Supplementary Method). This allows us to compare our analyzed boron isotope records to the constructed proxy record of ocean pH, carbon cycle disturbance, accounting for rapidly changing environmental conditions. To further test whether ocean acidification was a likely environmental stressor both at the beginning and during the T-OAE, we finally compare our boron proxy record of ocean pH to paleoecological patterns established for the Spanish and Portuguese sections[5,53,54], including changes in species diversity and shell size of benthic macroinvertebrate assemblages.

## Results and discussion
### Carbon, oxygen and boron isotope compositions in different carbonate components

Micrite ($n = 20$), brachiopod ($n = 21$) and bivalve ($n = 13$) samples from Spain, and micrite ($n = 20$) and brachiopod samples from Portugal ($n = 5$) selected for this study are from the Pliensbachian-Toarcian time interval

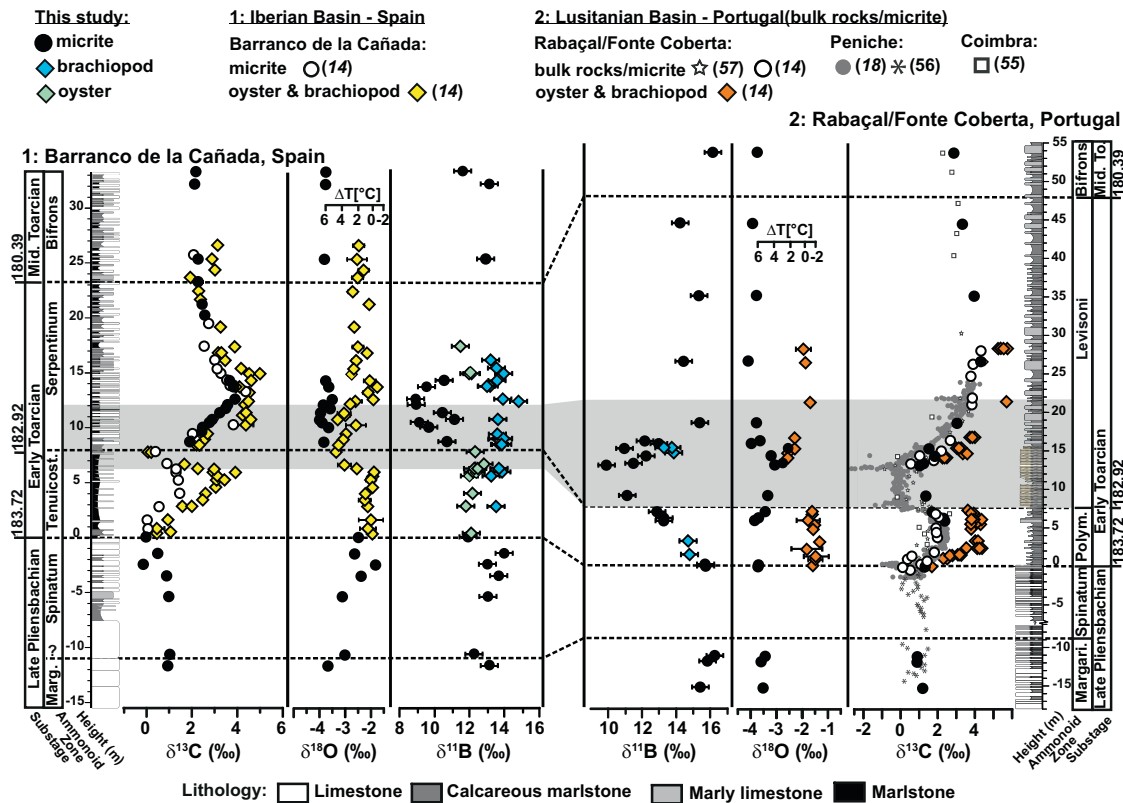

**Fig. 2 | Stratigraphy and isotope geochemistry of two Pliensbachian-Toarcian successions.** Boron isotope ($\delta^{11}$B) record (propagated uncertainty given as 2sf) for micrites, brachiopods and bivalves from Spain and Portugal (Methods). Cumulative carbon ($\delta^{13}$C) and oxygen ($\delta^{18}$O) isotope data (2sd$_{mean}$ uncertainty) from bivalves and brachiopods (combined under one symbol) and bulk rock from the two sections[14]. For comparison, micrite and bulk rock carbonate carbon isotope data from the Iberian (Spain[14]) and the Lusitanian Basin (Portugal[14,18,55–57]) are plotted. The shaded colored area represents the extent of the chemostratigraphically (negative CIE) defined T-OAE interval[14,54]. Lithological log of Spain modified after[79] and for Portugal after[80,81]. Ammonite biostratigraphic zones for Submediterranean (Spain) and Mediterranean provinces (Portugal) are from[82]. Ages are given in million years (Ma) according to Geological Time Scale 2016[83].

and include the T-OAE. The number of samples is constrained by our strict preservation criteria (Methods). All data are provided in the Supplementary Tables 1 to 4.

The $\delta^{13}$C values of brachiopods and bivalves show a distinct trend of rising values starting in the lowest Toarcian horizon (Fig. 2) interpreted to be the recovery from a negative CIE at the Pliensbachian-Toarcian (Pl-To) boundary[14]. A sharp decrease in $\delta^{13}$C values, defining the beginning of the T-OAE, is followed by a recovery towards higher values either at (in Spain) or after (in Portugal) the boundary of the Tenuicostatum (=Polymorphum) and Serpentinum (=Levisoni) zones. In the same interval, the $\delta^{18}$O values are initially stable but decrease at the onset of the T-OAE negative CIE. These $\delta^{13}$C and $\delta^{18}$O isotope values follow previously documented trends[14].

The $\delta^{11}$B values recorded by the brachiopods remain relatively stable through the early Toarcian and during the T-OAE with values around +13.7 ± 0.8 ‰ (2sd, $n$ = 21) in Spain and 14.1 ± 1.3 ‰ (2sd, $n$ = 5) in Portugal. This stability of $\delta^{11}$B values is observed both within and among the different genera of the rhynchonellid brachiopods. The bivalves (Spain) also show stable $\delta^{11}$B values with an average of +12.3 ± 0.7 ‰ (2sd, $n$ = 13), but with a data gap (lack of suitable material that passed our screening protocol) in the early Serpentinum Zone, and thus for most of the T-OAE (Fig. 2 and Supplementary Table 1, for sample selection see Methods).

The $\delta^{13}$C values from the micrite samples also trace the carbon isotope trends characteristic of the late Pliensbachian to middle Toarcian (Fig. 2) and concur with published data from brachiopods, bivalves and bulk rocks for each basin[4,14,18,55–57], arguing for a good preservation. While the $\delta^{18}$O$_{micrite}$ values in both sections are lighter than the comparable $\delta^{18}$O$_{fossil}$ values (see Supplementary Tables 1 and 2), they range from −4.1 to −1.8 ‰ for Spain and −4.3 to −1.8 ‰ for Portugal, indicating no major deep burial alteration during lithification (Method). The values overlap with published whole rock

data from the Iberian Basin[4] and are consistent with published data for the Lusitanian Basin[55].

The boron isotope record for the late Pliensbachian to middle Toarcian begins with constant $\delta^{11}$B$_{micrite}$ values in both sections (Fig. 2 and Supplementary Table 2), but with values for Spain (Pliensbachian to Pl-To boundary with +13.0 ± 1.5 ‰ (2sd, $n$ = 7)) slightly lower than for Portugal (Pliensbachian to lowest Toarcian with +15.8 ± 0.6 ‰ (2sd, $n$ = 5)). The $\delta^{11}$B$_{micrite}$ values decrease to +8.9 ± 0.2 ‰ (2sd) for Spain and +9.9 ± 0.2 ‰ (2sd) for Portugal at the time of the negative CIE. This decrease in $\delta^{11}$B values of 4 to 6 ‰ is transient and reaches its minimum in Spain at the end of the T-OAE, stratigraphically above the most negative $\delta^{13}$C values. The exact timing of the recovery cannot be determined due to a data gap (lack of suitable material that passed our screening protocol) in the upper Serpentinum Zone. In the correlative strata from Portugal, the return to high $\delta^{11}$B values is within the T-OAE interval and coincides within sampling resolution with the CIE. In both successions, $\delta^{11}$B values have recovered to heavier values (12.6 ± 1.6 ‰ 2sd, $n$ = 3 for Spain and 15.1 ± 1.5 ‰ 2sd, $n$ = 5 for Portugal) by the early and middle Toarcian as recorded prior to the T-OAE. The boron isotope pattern cuts across primary lithological boundaries, including limestones, marly limestones and marls (Fig. 2 and Supplementary Note 1), implying that the $\delta^{11}$B values are therefore both facies- and fabric-independent, as would be expected for a primary signal.

## Modelling approach

The COPSE model[51,52] tracks changes in the size and composition of biogeochemical reservoirs in the ocean and atmosphere over Phanerozoic time as a function of various time-dependent forcings and internal feedbacks. The purpose of the modelling exercise was to use greenhouse gas inputs to reproduce the $\delta^{13}$C values, then examine the impact of these inputs upon

**Fig. 3 | Model-data comparison. a** Compares the measured $\delta^{13}C$ data (yellow diamond[14]) with the model's estimate for the surface ocean carbonate carbon burial flux (solid blue line). **b** Compares the measured $\delta^{11}B$ values (micrite and brachiopods) with model estimates of the seawater average (solid blue line) and that associated with marine carbonate (solid purple line). **c–f** Compare the pH estimates derived from direct proxy inversion (symbols) with the model estimate for surface seawater pH (solid black line). Black and white circles show micrite pH data (**c** using the Klochko et al.[64] formula), other symbols and shades show brachiopod data (**d-f** equations are from Lécuyer et al.[34], Penman et al.[35] and Jurikova et al.[33]). Proxy inversion estimates are a function of the model $pK_B$ estimate (dashed black line), with the exception of that of Lécuyer et al.[34], for which it was held constant at $pK_B$ = 8.9 (Supplementary Method).

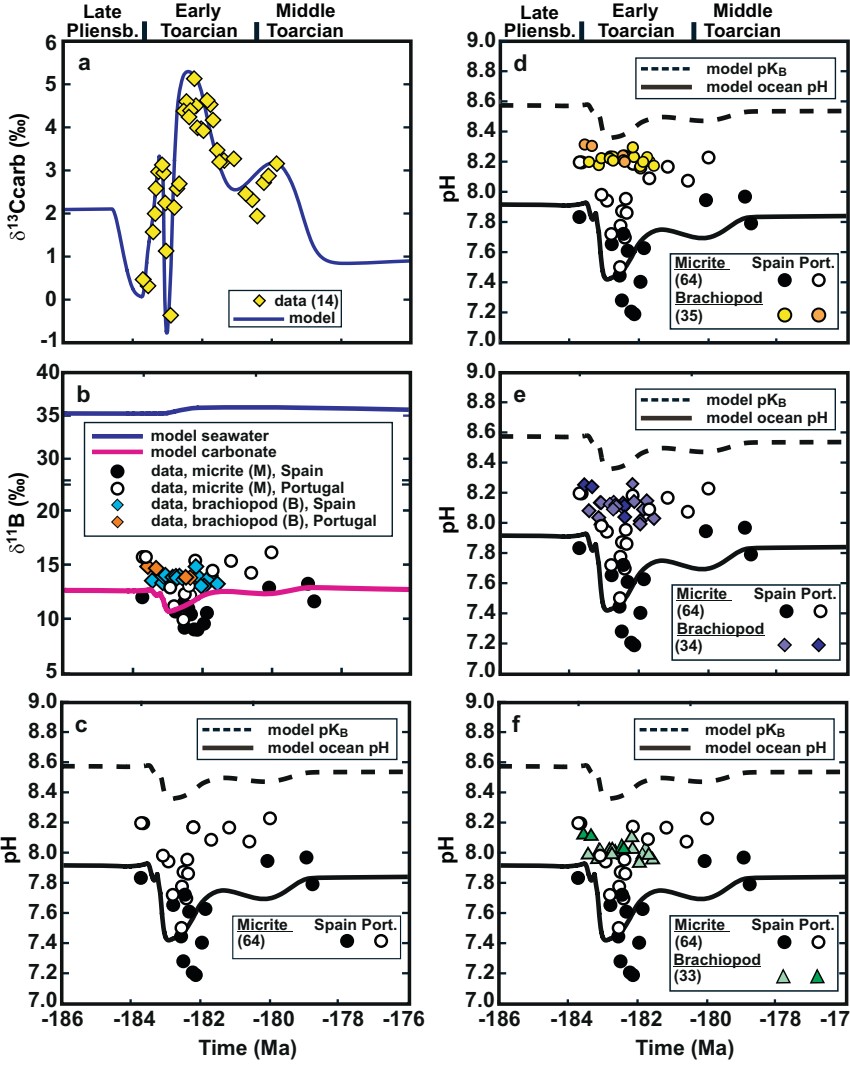

ocean pH and the boron cycle. To this end, COPSE was supplemented with an isotopically weighted mass balance model of the global boron cycle[58], as well as the CO2SYS model of the CO₂ system[59] to estimate surface ocean pH, and explicit pH-sensitive boron speciation functions[60]. The magnitude of greenhouse gas input fluxes of CO₂ and CH₄ were adjusted within reasonable bounds in order to cause the model to reproduce the previously published[14] $\delta^{13}C$ dataset (Fig. 3a). The greenhouse gas inputs are given in Supplementary Fig. 4, with Supplementary Figs. 5 to 9 showing example simulations with different input forcing isotopic compositions. There is no carbonate saturation dependence in marine carbonate carbon burial, because the system remains supersaturated (for further information on model inputs, see Supplementary Method and Tables 6 and 7). Isotopic compositions of inputs were $\delta^{13}C_{LIP}$ = 0 ‰ for large igneous province CO₂ (but other input values are also examined, see Supplementary Figs. 4 to 9) and $\delta^{13}C_{CH4}$ = −60 ‰ for clathrate methane[61]. Model ocean-atmosphere CO₂, temperature, total alkalinity (carbonate plus borate), surface phosphate concentration and boron reservoir sizes were used in conjunction with CO2SYS to estimate surface ocean pH. The pH estimate and associated model quantities were then used as inputs to Rae's[60] function, allowing the calculation of the dissociation constant of boric acid ($pK_B$) and carbonate $\delta^{11}B$ values (Fig. 3b, c). Temperature estimates from the model were compared to those derived from direct inversion of $\delta^{18}O$ values.

The combined input of isotopically neutral CO₂ ($\delta^{13}C_{LIP}$ = 0 ‰) and isotopically negative CH₄ was determined to be the best way to reproduce the $\delta^{13}C$ excursion[14] from early Toarcian to middle Toarcian time (see

Fig. 3a and Supplementary Figs. 4 to 9). The overall positive $\delta^{13}C$ excursion is attributable solely to model feedbacks. The greenhouse gas input results in increased CO₂ levels and temperature, leading to a weathering increase that elevates nutrient input to the oceans, raising productivity and marine organic carbon burial. This leads to an increase in the organic fraction of total carbon burial, which is responsible for the positive $\delta^{13}C$ excursion. The predicted lowering of pH persists through the positive $\delta^{13}C$ excursion. These changes, together with changes in the boron reservoir size (particularly connected to the increased carbonate carbon burial flux), is sufficient for the predicted borate-carbonate isotopic composition to roughly match the $\delta^{11}B_{micrite}$ pattern. The transient, weathering-induced increase in the marine P reservoir is sustained by anoxia (which has a negative impact on Fe-adsorbed phosphate burial), prolonging the OAE. The CO₂ sink provided by the elevated silicate weathering flux eventually brings the system back to pre-perturbation levels.

## Brachiopod and bivalve isotopic values and response to pH change

The $\delta^{11}B$ values in our rhynchonellid brachiopod record are stable in both Barranco de la Cañada, Spain and in Rabaçal/Fonte Coberta, Portugal and show no systematic changes across the T-OAE that could indicate a change in ocean pH. This is in notable contrast to the published B isotope record (+13.7 ± 1.4 ‰ (2sd, $n$ = 34) derived from diverse brachiopods (Rhynchonellida, Terebratulida and Spiriferida) across the same time interval from Peniche, Portugal (116 km SW from the Rabaçal/Fonte Coberta section),

which has been interpreted to document a total negative shift in ocean pH of about 0.5 units[28]. While the $\delta^{11}B$ values before and after the T-OAE are broadly similar (around 14 ‰) in both our own data and in the published record[28], the values from Peniche are scattered with a strong point to point fluctuation (12.1 to 15.6 ‰, $n = 42$) just prior to the onset of the T-OAE and limited data points from the T-OAE[28].

Using our model estimates for ocean pH, $pK_B$ and $\delta^{11}B_{SW}$ (average 35 ‰), the $\delta^{11}B_{brachiopod}$-ocean pH calibration established by Penman et al.[35] would place our brachiopod data in a pH range from about 8.15 to 8.30 across the T-OAE. The calibration established by Lécuyer et al.[34], which was used by Müller et al.[28] for the brachiopods from Peniche, would yield a pH scatter of 8.0 to 8.25 ($pK_B$ 8.9 = 0 °C) and the calibration by Jurikova et al.[33] of about 8.35 to 8.75 (Fig. 3d–f). Regardless of which pH calibration experiment is used, all calibrations reflect the low scatter of our $\delta^{11}B$ record, and no clear trend in ocean pH during the early Toarcian and during the T-OAE can be detected.

The discrepancy between our own brachiopod B isotope record and the published record[28] might be explained by differential preservation of the brachiopod samples. Even in the apparently well-preserved material from Peniche, there are some high Mn and Mg concentrations and high aluminum mass fractions of up to 1 wt.-%[28]. These values are suggestive of contamination by a Mg-rich clay, particularly in the dysoxic interval broadly equivalent to the T-OAE[32]. By contrast, the Al mass fraction in the brachiopods analyzed for this study is always below 70 μg g$^{-1}$ (0.26 mmol mol$^{-1}$ Al/Ca Supplementary Table 3). In modern brachiopods, substantial primary enrichments of aluminum are unknown[62,63] and mass fractions are typically reported to be below 0.24 mmol mol$^{-1}$. Extending this observation, once the $\delta^{11}B$ values are corrected for the effects of clay contamination in the Peniche samples, the trend to lower $\delta^{11}B$ values is no longer apparent[32].

An absence of any trend is also apparent when the data are not corrected but restricted to only those samples with Al/Ca ratios below 1000 μmol mol$^{-1}$ (about 270 μg g$^{-1}$ Al). After this screening procedure, the remaining $\delta^{11}B_{brachiopod}$ values from Peniche (13.1 to 15.6 ‰, $n = 18$) indeed broadly reproduce our brachiopod values (13.0 to 14.8 ‰, $n = 21$ for Spain and 13.3 to 14.8 ‰, $n = 5$ for Portugal). This underlines the importance of scrupulous screening of all sampled materials for contamination (Methods).

Fossil brachiopods are often used for geochemical analyses as they are considered to preserve their original skeletal chemistry in their secondary shell layer despite diagenetic processes, owing to the robustness of their low-magnesium calcite shell against such influences[14]. Studies addressing brachiopod $\delta^{11}B$ values as pH tracers have shown that the calcite of some modern terebratulid brachiopod shells records changes in $\delta^{11}B$ values with changing pH conditions in the ocean[33–35]. They all observed a pH sensitivity for $\delta^{11}B$ incorporation, but reduced sensitivity compared to the empirical aqueous fractionation in marine carbonates[64], indicating a degree of regulation of the pH of their internal calcifying fluids. In addition to this vital effect changing the $\delta^{11}B$ signature of the seawater in the shell, considerable species-specific $\delta^{11}B$-pH relationships were also observed (Fig. 3 and Supplementary Figs. 4 to 6). This suggests that using brachiopod $\delta^{11}B$ values to reconstruct changes in ocean pH conditions would only provide directly quantifiable results if single species were used, or by cross-calibrating coeval species, and only then for terebratulid species with some modern representatives. Various studies on the effects of ocean acidification on the shell growth of living and recent brachiopods have shown that brachiopods are generally resistant to lower pH values, have physiological buffering capacities for calcification and create suitable conditions at the site of calcification over a range of pH conditions[65,66]. The stable $\delta^{11}B$ values shown by the brachiopods, which differ from our B isotope micrite reconstructions and modelling, suggest that these particular Jurassic taxa were able to physiologically buffer changes in ocean pH and are hence of limited use for the interrogation of pH changes in Earth history.

Most importantly, we have chosen to interrogate the record of *Soaresirhynchia* and other rhynchonellid brachiopods precisely because these taxa survived the T-OAE event so can be analyzed to provide a continuous record before, during and after the T-OAE. *Soaresirhynchia bouchardi* was

an opportunistic brachiopod species that dominates late T-OAE records[5,53,54]. This suggests a notable resilience to the kill mechanisms of the extinction, which might be imparted by the ability to buffer internal pH and also survive the low oxygen conditions that occurred co-incident with the T-OAE. This has been well-documented for other extinction events, such as the end-Permian, where biotas already adapted to low oxygen, such as those dominated by the bivalve *Claraia*, preferentially survived the extinction[67]. Indeed, we note that the T-OAE extinction was selective against physiologically-unbuffered organisms, i.e., calcifiers and hypercalcifiers such as corals and bivalves[25,29].

There is the data gap across the T-OAE for the oyster samples (Spain), but the stable, rather than fluctuating, $\delta^{11}B$ values are consistent with our brachiopod B isotope record (Fig. 2 and Supplementary Table 1). This again contrasts notably with the published brachiopod data from Peniche, which show the sharpest decline in $\delta^{11}B$ values by ~2 ‰ from the Pl-To boundary with the lowest value in the Polymorphum Zone[28], and with our model and micrite record.

As in brachiopods, oysters construct their outer shells of low-magnesium calcite, using the extrapallial fluid for calcification with the ability of regulating the calcifying fluid and maintaining stable pH conditions[37]. Experimental studies on the response of the B isotope composition of the living American oyster *Crassostrea virginica* to changing ocean pH conditions[38,39] support the idea that oysters exert moderate-to-strong control over the pH of their calcifying fluid, keeping the pH of the extrapallial fluid elevated relative to the seawater pH under acidified conditions. That the regulation of calcification site pH poses a limitation for the use of boron isotopes to reconstruct seawater pH from oysters was recently confirmed[40]. There is no B isotope-ocean pH calibration for oysters.

## Micrite B isotope trends and model constraints on ocean pH

While micrite might appear far from ideal to serve as a reliable $\delta^{11}B$-ocean pH archive, reproducing identical, facies- and fabric-independent patterns in two disparate sections across the same time interval argues for an original and inter-regional record. Alternative explanations for the change in the B isotope composition, such as diagenesis or contamination, can be ruled out by optical and geochemical screening (Method). Another explanation might be that the successions saw a shift in the composition of organisms contributing to micrite formation. We see, however, no evidence for a change in the inventory of benthic vs pelagic carbonate contribution to the micrite record. Similarly, there is no evidence that environmental processes (changes in temperature, salinity, bathymetry, weathering) other than acidification could have been the main cause of the temporary decrease in $\delta^{11}B$ values (Supplementary Discussion).

Accepting the Pliensbachian and Toarcian micritic carbonates as a proxy archive for ocean pH changes, the transient decline in $\delta^{11}B$ values supports the hypothesis of ocean acidification during the T-OAE (Fig. 2). This ocean acidification event, however, is transient, and the $\delta^{11}B$ values—and by implication ocean pH—in Portugal recover within the T-OAE interval towards more alkaline values. In Spain, the acidification event seems to persist through the T-OAE and recovery to more alkaline values occur in the aftermath of the T-OAE and the middle Toarcian. The apparent discrepancy may reflect differences in sedimentation rate, and cryptic hiatuses between the two localities. Also, within existing stratigraphic control, age models are not highly resolved and are constantly being refined. Nevertheless, both sets of data show a relationship between $\delta^{11}B$ values and the T-OAE CIE.

According to the model, the inferred $CO_2$ and $CH_4$ release events caused by Karoo and Ferrar volcanism, which reproduce the $\delta^{13}C$ curve[14], lead to continued ocean acidification, which in turn causes the model to reproduce the pattern of micritic $\delta^{11}B$ values (Fig. 3). In our model, the onset of the decline in ocean pH (from 7.9) starts at the Pl-To boundary and is linked to early magmatism of the Karoo and Ferrar LIPs[6–9] at about 183.5 Ma. This early decline in ocean pH continues into the early Toarcian (during the Polymorphum/Tenuicostatum Zone), prior to the onset of the negative CIE of the T-OAE, but is only considered to amount to 0.1 pH units

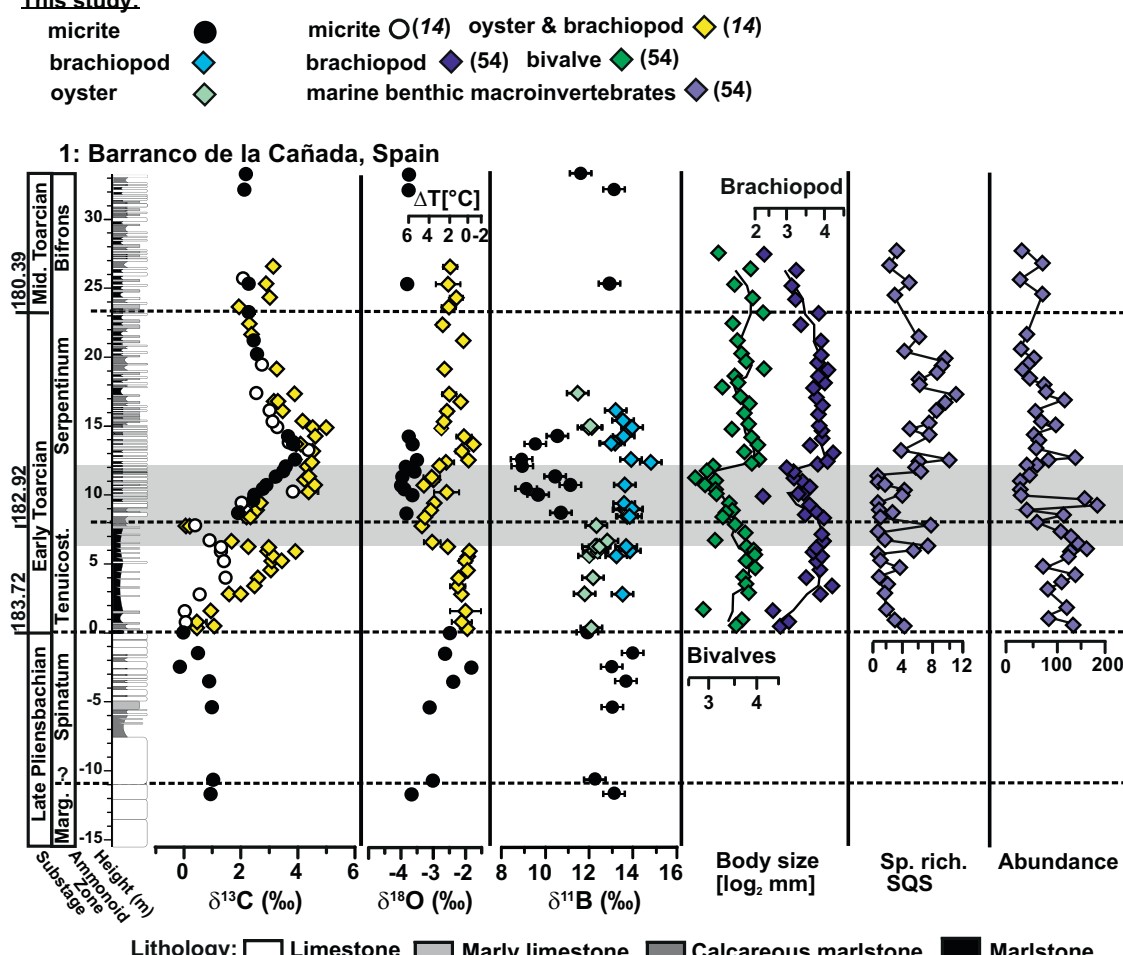

**Fig. 4 | Isotope geochemistry and paleoecology at Barranco de la Cañada, Spain.** Isotope geochemical data ($\delta^{13}C$, $\delta^{18}O$ and $\delta^{11}B$), mean shell sizes ($\log_2$ mm) of brachiopod and bivalve assemblages[5], standardized species richness (Shareholder Quorum Subsampling [SQS] diversity) and abundance (absolute number) of specimens in samples (standardized by sample weight) of the Barranco de la Cañada succession. Selected biodiversity metrics are based on the taxonomic composition of faunal samples[54].

down to 7.8 for the global ocean (Fig. 3c). Such a decline in ocean pH is compatible with the $\delta^{11}B_{micrite}$ values in Spain and Portugal, but cannot be unambiguously deduced due to the sparse data coverage in the Polymorphum (Tenuicostatum equivalent) Zone. Ocean pH continues to decline to a pH value of just above 7.4 for the global ocean in the T-OAE. This drop in ocean pH coincides with the largest influx of LIP $CO_2$ and the suggested astronomically paced, widespread release of methane from various sources[10,15–17], which in turn drives the decrease in $\delta^{13}C$ values during the T-OAE. The minimum in modelled global ocean pH with a decrease in pH of close to 0.5 units occurs synchronously with the T-OAE, i.e., overlapping with the most negative C isotope values and with the decreasing trend of the $\delta^{11}B_{micrite}$ values in both sections. The return to global ocean pH values similar to pre-Toarcian conditions is gradual and continues beyond the T-OAE during the Serpentinum Zone.

Within the available data and age model resolution, the B isotope patterns for Spain and Portugal broadly follow the modelled ocean pH pattern. However, the pattern for Portugal suggests a shorter-lived acidification event with a recovery within the T-OAE (Figs. 2 and 3c). The reasons for this apparent decoupling in recovery are not clear. Differences in bathymetry, salinity and water temperature between the two sites cannot explain the divergent pattern, but varying weathering intensities with different buffering capacities of the two sites coupled with basin restrictions might offer possible mechanisms (see Supplementary Discussion). Both sections are located in different basins on opposite margins of the Iberian Massif, which could lead to regional differences in weathering intensity. Increased continental weathering during the T-OAE with locally varying rates has already been proposed, with a fivefold increase observed for the Lusitanian Basin in Portugal based on calcium isotopes[68]. The calcium and carbon isotope excursions show a temporal synchronicity in the T-OAE[68], which also is reflected in our boron isotope excursion, potentially indicating pH regulating in the ocean. Although weathering could explain the divergent B isotope excursion, further investigations are needed as corresponding weathering rates are not known for the Iberian Basin in Spain.

**Ocean acidification and faunal response**

The decline in $\delta^{11}B$ values in micrite in both the Spanish and Portuguese sections and the onset of low pH conditions in the model coincide with the chemostratigraphically defined onset of the T-OAE[14] (Figs. 2 and 3c), which marks the main extinction phase with pronounced assemblage changes in multiple fossil groups e.g.[2,4,54]. This suggests that the drop in ocean pH might have functioned as a kill mechanism, together with global warming, ocean deoxygenation and/or changes in nutrient cycling, collectively termed the deadly quartet of coupled stressors which often occur during mass extinctions, including the T-OAE[1,53,69].

High resolution diversity trajectories, extinction patterns and changes in ecological structure and shell size of benthic macroinvertebrate assemblages established for the sections in Spain and Portugal[5,53,54] show a complete turnover of brachiopod species in the lower part of the T-OAE,

followed by the strong predominance of an opportunistic brachiopod species, *Soaresirhynchia bouchardi*, in the upper part of the T-OAE (Supplementary Table 1). In the Spanish section (Fig. 4), multiple faunal indicators of stress (raised extinction intensity of brachiopod species, decreasing and fluctuating taxonomic and functional diversity, decreasing macrobenthic biomass) correlate with an increase in water temperatures in the lower part of the T-OAE, as derived from the oxygen isotope record of shells[54].

The lack of boron isotope data from micrite in the lower part of the T-OAE does not allow for any statistical correlation tests of fossil data with geochemical proxy data. However, the inferred low pH coincides with the final disappearance of the existing brachiopod species in both Spain and Portugal, and low pH in the upper part of the T-OAE is accompanied by small shell size, low species richness and overall ecological instability[5,53,54] (as e.g., indicated by the dominance of the opportunist *S. bouchardi* along with other opportunistic species) (Fig. 4). In the middle part of the T-OAE in Portugal, high water temperatures[14] and low pH coincide, and again the fauna is characterised by a low-diversity disaster fauna dominated by *Soaresirhynchia*.

These observations might suggest that ocean acidification could be an additional environmental stressor to the temperature increase both at the beginning and during the T-OAE. By contrast, at the end of the T-OAE, rapid recovery of shell size, diversity, and ecological structure in Spain is synchronous with decreasing temperatures, whereas pH derived from carbon and boron isotope estimates remained low.

We conclude that ocean acidification may have played an additional role in the faunal turnover around the Tenuicostatum-Serpentinum zone boundary and the establishment of a stressed low-diversity fauna dominated by the opportunistic and therefore potentially physiologically resilient *Soaresirhynchia*, but that the recovery of communities appears to be independent of pH changes.

### Implications for capturing ocean pH changes in deep time

Our high-resolution B isotope record of different carbonate components provides critical new insights into how to best capture deep time oceanic pH changes. First, meticulous screening for late diagenetic overprinting, geochemical preservation and mineral contamination of samples is vital in order to correctly recover accurate and meaningful B isotope values. Second, our data suggest that some rhynchonellid brachiopods, particularly those that form part of opportunistic faunas which may indicate physiological resilience to extinction kill mechanisms, and arguably oysters, may be poor targets for the interrogation of pH changes in deep time. This may be due to their ability to physiologically buffer changes in ocean pH. Yet, micrite B isotope values are inferred to more faithfully track relative ocean pH changes, and excursions that indicate temporary ocean acidification have been detected at two independent localities. These events may, however, have played only a minor role in faunal turnover at the T-OAE, and community recovery appears to be independent of such ocean pH change.

We conclude that interrogation of micrite using B isotope values might better capture changes in ocean pH in deep time. While the absence of a usable $\delta^{11}$B-ocean pH calibration for micrite in the modern makes an accurate inference of actual values of ocean pH in deep time not yet possible, we can nonetheless show here that B isotope values derived from micrite can capture trends in ocean pH changes.

## Methods
### Samples
Marine carbonates and biogenic material from carbonate ramp paleoenvironments were collected from two different locations in the western Tethys: Rabaçal/Fonte Coberta (Portugal; 40°03'08.0"N 8°27'30.5"W) representing the Lusitanian Basin, and Barranco de la Cañada (Spain; 40°23'53.4"N 1°30'07.4"W) for the Iberian Basin (Supplementary Note 1). Both sections are about 600 km apart. Samples were collected at distances varying from 0.1 to 11 m scale intervals, covering the upper Pliensbachian to the middle Toarcian, including the Pliensbachian – Toarcian boundary (Pl-To) and the T-OAE. In total, 40 micritic carbonate rock, 26 brachiopod

(Rhynchonellida) and 13 bivalve (oyster *Gryphaea*) samples were analyzed. For brachiopods and bivalves, assessment of preservation, element mass fraction, and oxygen and carbon isotopes are documented[14] and summarized below (Methods). Since the assemblage changes were too severe across the T-OAE at both sample locations, we were not able to sample a record based only on a single species of brachiopod. We nevertheless used only rhynchonellid brachiopods to ensure the best possible results. Fossil material from Fonte Coberta / Rabaçal is archived at the Museum für Naturkunde, Berlin, Germany (samples MB.B.10843-10912 for brachiopods and MB.B.20325-20346 for bivalves). Shell fragments from Barranco de la Cañada are stored at the Museu de Ciencias Naturales, Zaragoza, Spain (samples MPZ 2019/415-571).

### Sample preparation—micritic material
Macroscopically screened micritic carbonate samples with no obvious signs of secondary alteration were cut into thin (mm) slices with a diamond saw. For SEM and CL imaging, small pieces of the carbonate slices were chipped from each sample, embedded in epoxy resin, ground and polished with a final grain size of 0.05 μm. Each slice was examined under a binocular microscope and carefully broken into small pieces with a preparation needle. During sampling for elemental and isotope analysis, only the homogeneous micrite matrix was picked and any shells of macroinvertebrates and areas of burrow traces and veins were avoided. The selected chips were manually ground with an agate mortar, but not too fine to prevent the abrasion of any clay minerals. The powdered sample material was washed with ultrapure water (Milli-Q) to remove any water-soluble fraction, dried, and dissolved in 1 M HCl to leach 80 % of the carbonate fraction. The leachate was used for further analyses. To determine the weight percentage of the detrital fragments in the micrite samples, 100 mg of the sample material was dissolved with HCl and the residue was washed with ultrapure water. The residue was evaporated at 90 °C and weighed (Supplementary Table 4).

### Sample preparation—shell material
Fossil preparation was conducted at the Penryn Campus, Environment and Sustainability Institute, University of Exeter and is described in detail[14]. Initially, the surface of all brachiopod and bivalve shell material was cleaned from sediments and altered rinds. If present, the primary shell layer of the brachiopods was removed using a preparation needle, scalpel or hand-held drill with diamond coated drill bit. In general, shell material for analyses was extracted as sheaths of multiple shell layers with a preparation needle. For specimens with a dense nature of the material (mostly for *Gryphaea*), samples were taken with a scalpel or hand-held drill using a diamond coated drill bit of ca 1 mm diameter. Calcite splinters of ca 1 mm width were prepared for SEM imaging using an FEI Quanta 650 Field Emission Gun Scanning Electron Microscope (FEG SEM). The size of sampled shell material for C and O isotope analyses ranged typically between 1 and 3 mg. For element analyses 220 to 820 μg of sample material was dissolved in 2% v/v $HNO_3$ with a ratio of 15 mL per mg fossil calcite resulting in a dilution to ca 25 μg g$^{-1}$ Ca. For B isotope analysis, the shell material (typically around 10 mg) was cleaned three times with ultrapure water (Milli-Q) and ethanol and dissolved with 1 M HCl for a maximum of 24 h until the reaction had ceased.

### Optical analyses of the microstructure
All optical analyses of the carbonate rock microstructure were done in the laboratories of the research group Petrology of the Ocean Crust, University of Bremen. Cathodoluminescence (CL) images were taken using the Technosyn Cold Cathode Luminescence Model 8200 Mk II and are used to observe the internal structure and to assess the degree of diagenetic alteration of the carbonates. Further assessments of the carbonate rock microstructure were performed by Scanning Electron Microscopy (SEM), using the field emission electron microscope SUPRA 40 (Zeiss) coupled with an EDX detector XFlash 6|30 (Bruker). A detailed study of fossil shell preservation employing optical analyses (binocular microscope, FEG SEM) was

performed at the Environment and Sustainability Institute, University of Exeter[14].

## Elemental analysis

Element mass fractions for the micrite (Supplementary Table 4) were determined by inductively coupled plasma optical emission spectrometry (ICP-OES) on an Agilent 700 at the Sediment Geochemistry Group at MARUM, University of Bremen (for the samples from Portugal) and on the Thermo Scientific iCAP 7400 at the Marine Geochemistry Section, Alfred Wegener Institute, Helmholtz Centre for Polar and Marine Research (AWI) (Spain samples). High purity certified multi-element standards (GSJ CRM JCt-1 coral and JCp-1 giant clam) were used for element-specific instrumental calibration. Based on repeated measurements of the certified reference material, the relative uncertainties (2rsd) in both laboratories were 5% for all elemental mass fraction determinations.

The shell samples were analyzed for element/Ca ratios using an Agilent 5110 VDV ICP-OES with Seaspray U-series glass nebulizer and double pass cyclonic spray chamber at the University of Exeter. Analytical calibration was done by quality control solutions (BCQC & BCQ2) and reference materials (JLs-1 limestone and UN AK carbonate). Reproducibility for element/Ca ratios are within 1 % (2rsd) for all reference materials. Absolute Ca mass fractions calculated for fossils reproduced to better than 0.6 wt.-% (2sd). Analytical details and results for all measured standard solutions are listed in ref. 14. Element mass factions and element/Ca ratios for fossils analyzed in this study are listed in Supplementary Table 1 and 3.

Boron mass fractions in the micritic (Supplementary Table 4) and fossil (Supplementary Table 1) material were determined during the course of boron isotope ratio measurements, using a ThermoFisher Scientific Neptune Plus Multicollector-inductively coupled plasma-mass spectrometer (MC-ICP-MS) at the University of Bremen, with an uncertainty (intermediate precision) of 9 % (2rsd) based on long term (period of 3 years) analyses of NIST SRM 951 ($n = 108$). The small sample size results in a weighing uncertainty of about 10 %.

## Carbon and oxygen isotope analysis

Carbon and oxygen isotope ratios on micritic material were analyzed with a Thermo Finnigan MAT 252 gas isotope mass spectrometer coupled to an automated carbonate preparation device (Kiel III) at the Stable Isotope Laboratory at MARUM, University of Bremen, Germany. All carbon and oxygen isotope values (Supplementary Table 2) are quoted in the conventional δ per mil (‰) notation relative to VPDB. Measurements were calibrated against the house standard (SHK 2008, Solnhofen limestone), itself calibrated to RM 8544—NBS 19. Repeatability of replicate analyses of the house standard was better than ±0.1 ‰ (2sd) for $\delta^{13}C$ and $\delta^{18}O$.

Shell samples were analyzed for $\delta^{13}C$ and $\delta^{18}O$ values as well as carbonate content using a SerCon 20-22 Gas Source Isotope Ratio Mass Spectrometer (GS-IRMS) in continuous flow mode at the Environment and Sustainability Institute, University of Exeter. The results on the isotope data are reported in the conventional δ per mil (‰) notation relative to VPDB, measured standardized against the in-house standard CAR (Carrara Marble) and NCA (Namibia Carbonatite) that are calibrated against the RM 8543—NBS 18, CO-8 and LSVEC. The 2sd repeatability was found to be better than ±0.09 ‰ for $\delta^{13}C$ and ±0.35 ‰ for $\delta^{18}O$. The carbonate mass fraction was calculated against the carbonate mass fraction in CAR, which was assumed to be pure $CaCO_3$ (44.0 wt.-% $CO_2$). Details on sample preparation, measurement techniques and analytical uncertainties are given in ref. 14. Analytical results for isotopic composition and $CaCO_3$ content on fossil material for this study are listed in Supplementary Table 1.

## Boron isotope analysis

Boron isotope ratios of micrite and fossil material were analyzed in the Isotope Geochemistry Laboratory at the MARUM—Center for Marine Environmental Sciences, University of Bremen. The B purification of micrite and fossil shell materials was performed using a modification of the cation exchange method[70,71]. Bio-Spin ® columns were filled with 1 ml of

Bio-Rad resin AG 50WX8 (200-400 mesh size), cleaned with 6.2 M HCl, and conditioned with 0.02 M HCl. Sample aliquots containing about 250 ng B were dried in presence of mannitol, dissolved in 0.02 M HCl, placed on the resin and eluted with 0.02 M HCl. Boron recovery during the cation column separation was 99.8 %, and the procedural blank was less than 2 ng B and had no considerable influence on the isotopic composition and B mass fraction of the samples. Boron isotope ratios of all samples were analyzed with a ThermoScientific Neptune Plus MC-ICP-MS, using a stable introduction system (SIS) and a high-efficiency x-cone[72]. Isotope ratios were measured using the standard-sample-bracketing method with reference material NIST SRM 951 as the bracketing standard. Each sample was analyzed at least three times in blocks, and 2 % $HNO_3$ was used for baseline corrections. Isotope ratios are reported in the conventional $\delta^{11}B$ (‰) notation relative to NIST SRM 951. The accuracy and repeatability of the sample purification and measurement procedure was verified through various reference materials. The obtained results for these reference materials are within analytical uncertainty in agreement with the literature values. Repeated measurements of the in-house standard bottom seawater from SuSu Knolls (BSW) for the course of this project yielded a $\delta^{11}B$ value of 39.7 ± 0.4 ‰ (2sd, $n = 28$), which is in agreement with published data on modern seawater (39.61 ± 0.04 ‰ ($2sd_{mean}$)[73]). Boron isotope values for GSJ CRM JCt-1 (coral + 16.3 ± 0.2 ‰ (2sd, $n = 2$)) and for JCp-1 (giant clam + 24.5 ± 0.1 ‰ (2sd, $n = 1$)), are in agreement with published values from e.g.[74], (JCp-1 + 24.36 ± 0.45 ‰ (2s*) and JCt-1 + 16.39 ± 0.60 ‰ (2s*)). Long-term repeatability (intermediate precision) of NIST SRM 951 yields a $\delta^{11}B$ value of −0.1 ± 0.1 ‰ (2sd, $n = 37$; over a period of 3 years). The uncertainty of the sample material is reported as two standard deviation (2sd) measurement precision, based on multiple mass-spectrometer analyses (Supplementary Table 1 and 2). For a final uncertainty statement in the figures, the uncertainty of the reference material was propagated on the analytical uncertainty of the sample (2sd*f*; typically, less than 0.45 ‰).

## Potential influence of contamination and diagenesis on micrite

A major challenge in using Pliensbachian—Toarcian carbonate rocks as archives for ocean pH conditions is to identify and select high-quality samples in which primary B isotope signals are still preserved. To achieve this, the sample selection procedures that have been successfully applied on Neoproterozoic and Permo-Triassic carbonate rocks to reconstruct ocean pH variation[41,43,44] were used. Hence, samples selected for B isotope analyses were screened macroscopically (in the field and the laboratory), microscopically (scanning electron microscope) and geochemically (carbon and oxygen isotopes, trace elements).

Cathodoluminescence (CL) images indicated that none of our analyzed carbonate rock samples showed patterns of growth zones, discontinuities and peculiar cementation fabrics. The luminescence was according to their analyzed Mn and Fe mass fractions, ranging in color from yellowish to reddish, implying an incorporation during recrystallization. SEM images (backscatter and secondary electron images as well as element mappings) showed no signs of alteration and no dolomitization of the matrix (Supplementary Fig. 2). EDX screening provided a first overview on the type of detrital minerals (e.g., quartz, mica, feldspar, clay minerals, see Supplementary Discussion and Table 5).

Trace element analyses were performed on the micritic material for assessing the diagenetic overprint and potential contamination by detrital mineral dissolution (Supplementary Table 4). In the selected samples, Mn mass fraction is <270 µg g$^{-1}$, Sr mass fraction ranges from 210 to 638 µg g$^{-1}$, and the Mn/Sr ratio is low (<1 w/w), suggesting no influence of meteoric fluids. The boron mass fraction for the carbonate rocks is, on average, 1.5 µg g$^{-1}$ and ranges between 0.5 and 2.9 µg g$^{-1}$. Potential analyses of disseminated detrital material in the carbonates was also checked through elevated Al and Si mass fractions and showed either no or only minor dissemination of clay. In addition, there is no statistically significant relationship between e.g., aluminum and boron mass fractions (*p* value of 0.8 (Spain) and 0.2 (Portugal)) or boron isotope composition (*p* = 0.2). There appears to be a relationship between boron mass fraction and boron isotope

data, but this is due to the generally higher B mass fraction in the older Pliensbachian samples (Supplementary Table 4 and Fig. 3).

Post-depositional alteration during the conversion from unlithified lime mud into micrite, and especially meteoric diagenesis and recrystallization, is assumed to decrease the isotopic composition of oxygen, boron and carbon isotopes[56,57]. Ullmann and colleagues[14] published carbon and oxygen isotope values for the Spanish and Portuguese sections from well-preserved brachiopods and bivalves. Our $\delta^{13}C_{micrite}$ values overlap with the fossil values for both sections and show the distinct carbon isotope trend found for the Pliensbachian-Toarcian time interval, arguing for a good micrite preservation (Fig. 2). In addition, the $\delta^{13}C_{micrite}$ values for the Portuguese section are well aligned with published bulk rock and micrite data for the Lusitanian Basin[14,18,55–57]. For the Spanish section, the published $\delta^{13}C$ rock values show a wider scatter[4,14], with our micrite data lying amidst the published data. In contrast, the $\delta^{18}O_{micrite}$ data are on average 1 ‰ lighter than the $\delta^{18}O_{fossil}$ data, pointing to a potential diagenetic overprint. However, $\delta^{18}O_{micrite}$ values between −4.1 and −1.8 ‰ for Spain and −4.3 to −1.8 ‰ for Portugal, indicate no considerable deep burial alteration during lithification (Supplementary Table 1 and 2) and are consistent with published whole rock data from the Iberian Basin[4] and the Lusitanian Basin[55].

The boron isotope pattern cuts across primary lithological boundaries, including limestones, marly limestones and marls (Fig. 2 and Supplementary Note 1), implying that the $\delta^{11}B$ values are therefore both facies- and fabric-independent, as would be expected for a primary signal. The similarities in the $\delta^{11}B$ pattern in two comparable but independent transects from opposite margins around the Iberian Massif are a further indication of the preservation of a primary seawater isotope signature as e.g., late diagenetic features are expected to have high lateral variability and disrupt the uniform and consistent trends seen in both sections. If we assume that the boron isotope excursion is a diagenetic feature, then we would have to consider a diagenetic front that affects only about 7 m of the transect, roughly coincides with the T-OAE, cuts across different facies, and all this in marine carbonate sequences deposited in the NW Tethys in two different basins and at different water depths. Although such a scenario is not impossible, it is highly unlikely and we see no evidence for it.

## Fossil shell preservation

Rhynchonellid brachiopods from Spain are exceptionally well preserved, displaying the typical characteristics of minimal recrystallization like a silky reflection and slightly brownish semi-transparent colors when viewed under the binocular microscope. This allowed the extraction of shell fibres in well-defined packages of multiple layers along fibre surfaces. SEM imaging confirmed the preservation of the single-crystal character of the fibres with no indication of pervasive re-crystallisation. Material with e.g., large cement filling and clearly recrystallized calcite was not sampled for geochemical analysis. Shell preservation of bivalves (*Gryphaea*) from Spain is good, but optical examination revealed areas of shell calcite with incipient recrystallization up to the loss of the shell structure. However, these features were seldom pervasive and primary shell textures are preserved in nearly all studied shell fragments. This allowed sampling of visually best-preserved shell material with the typical stacks of thin, translucent sheaths of calcite layers for geochemical analysis[14].

While showing the same morphological characteristics as the brachiopods from Spain, the majority of the specimens from Portugal showed clear evidence for poor preservation and diagenetic overprints, such as fusion of shell fibres and neomorphic calcite during optical assessment. The same applies to the bivalves from Portugal[14]. Consequently, only five brachiopod specimens were selected for further investigation, but no bivalve could be considered.

In order to exclude altered fossil material effectively, common limits of good preservation of 0.1 mmol mol$^{-1}$ for Mn/Ca and 1.0 mmol mol$^{-1}$ for Fe/Ca ratios were adopted for the brachiopod and bivalve samples[14]. Aluminum is not commonly used to assess the preservation of the fossil material in terms of chemical composition, and since optical screening did not reveal any clay mineral content, aluminum was not analyzed in the original

dataset[14] (Supplementary Table 1). In view of the sometimes extremely high Al mass fractions of up to 1 wt.-% published for brachiopod samples from Peniche[28] and the associated potential contamination of the B isotope values by clay minerals, as discussed by Li et al.[32] and in the main text, the element/Ca ratio in representative brachiopod samples for which shell material was still available was re-analyzed (Supplementary Table 3). Samples were selected so that all important taxa were covered by multiple samples from both studied localities. None of the material re-analyzed here showed Al/Ca ratios approaching values as high as those reported from Peniche[28]. The Al mass fraction in our brachiopods from Spain and Portugal even for samples that would have been excluded on the basis of somewhat elevated Mn ($n = 2$) or elevated Fe ($n = 2$) mass fractions was always below 70 µg g$^{-1}$ and 0.26 mmol mol$^{-1}$ Al/Ca, respectively, which is near the limit of quantification for the method. The Rb/Ca ratio was always below detection limit. In modern brachiopods, significant primary enrichments of aluminum are unknown and mass fractions are typically reported[62,63] to be below 0.24 mmol mol$^{-1}$. Ratios of more than 1 mmol mol$^{-1}$ (up to 68 mmol mol$^{-1}$) Al/Ca, as reported in Müller et al.[28] more likely reflect partial leaching of clay contamination in fossil materials, which would be expected to variably affect different element/Ca and isotopic ratios.

## Reporting summary
Further information on research design is available in the Nature Portfolio Reporting Summary linked to this article.

## Data availability
The authors declare that all data supporting the results of this study are available in the main text or in the Supplementary Information including sampling area (including lithology) and the model description. All data are open accessed archived in the World Data Center PANGAEA—Data Publisher for Earth & Environmental Science at: https://doi.org/10.1594/PANGAEA.981213[75].

## Code availability
Full model code is available at https://doi.org/10.5281/zenodo.15699150[76] and https://github.com/richboyle111/COPSEBoron.

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

## Acknowledgements

We thank F. Lucassen (Faculty of Geosciences, University Bremen) for support during fieldwork and H. Kuhnert (MARUM, University of Bremen) for support with the carbon and oxygen isotope analyses. Thanks to the research groups Geodynamics of the Polar Regions, Petrology of the Ocean Crust, and Crystallography (Faculty of Geosciences, University of Bremen) for support with sample preparation, SEM and CL imaging and X-Ray Diffraction analyses, respectively. Thanks to the Sediment Geochemistry Group (MARUM, University of Bremen) and the Marine Geochemistry Section (AWI Bremerhaven) for the element analysis. We would like to thank Juan Carlos Garcia and the Direccion General de Cultura y Patrimonio (Gobierno de Aragon, Zaragoza) for authorizing our fieldwork and José Ignacio Canudo (Universidad de Zaragoza) for the loan of specimens. This study received funding from the German Research Foundation grant DFG KA 3192/4-1 and forms part of the Research Unit TERSANE (FOR 2332: Temperature related Stressors as a Unifying Principle in Ancient Extinctions). R.W. acknowledges support from NERC project NE/T008458/1.

## Author contributions

Conceptualisation and funding acquisition: S.A.K. and R.A.W. Software and Methodology: R.B. and T.M.L. Investigation: T.K., C.V.U., M.A., L.V.D., A.M., V.P. and R.B. Visualization: T.K., R.B. C.V.U. and S.A.K. Supervision: A.M., R.A.W., T.M.L. and S.A.K. Writing—original draft: T.K., R.B., and S.A.K. All authors contributed to the interpretation of the results and the preparation of the final manuscript.

## Funding

## Competing interests

The authors declare no competing interests.
