## [Transparent Peer Review file · Communications Earth & Environment]

Ocean acidification at the Toarcian Anoxic Event captured by boron isotopes in the lime mud record

Corresponding Author: Professor Simone Kasemann

Version 0:

Decision Letter:

Dear Professor Kasemann,

Your manuscript titled "Ocean acidification at the Toarcian Anoxic Event captured by boron isotopes in the lime mud record" has now been seen by 3 reviewers, and we include their comments at the end of this message. They find your work of interest, but some important points are raised. We are interested in the possibility of publishing your study in Communications Earth & Environment, but would like to consider your responses to these concerns and assess a revised manuscript before we make a final decision on publication.

We therefore invite you to revise and resubmit your manuscript, along with a point-by-point response that takes into account the points raised. Please highlight all changes in the manuscript text file.

In particular, please ensure that the revised manuscript meets the following editorial thresholds:

- 1) Provide form and sufficient evidence to demonstrate that bivalves and brachiopod taxa may have been able to buffer changes in ocean pH.
- 2) Fully justify your methodology in response to the criticisms raised by the reviewers. In particular, provide a clear explanation and/or evidence to justify your choice of B-proxy to study ocean acidification and your choice of model.

Please submit your point-by-point responses as a separate file, distinct from your cover letter where you can add responses to the Editors' comments that you do not want to be made available to the reviewers. Word files are preferred. We recommend that any figures, tables or graphs that are included in the response to reviewers are also included in the main article or Supplementary Information.

Please use the following link to submit your revised manuscript, point-by-point response to the referees' comments (which should be in a separate document to any cover letter), a tracked-changes version of the manuscript (as a PDF file) and the completed checklist:

Link Redacted

We hope to receive your revised paper within six weeks; please let us know if you aren't able to submit it within this time so that we can discuss how best to proceed. If we don't hear from you, and the revision process takes significantly longer, we may close your file. In this event, we will still be happy to reconsider your paper at a later date, as long as nothing similar has been accepted for publication at Communications Earth & Environment or published elsewhere in the meantime.

Please do not hesitate to contact us if you have any questions or would like to discuss these revisions further. We look

forward to seeing the revised manuscript and thank you for the opportunity to review your work.

Best regards,

Jun Shen
External Editor
Communications Earth & Environment

Carolina Ortiz Guerrero, PhD (she/her/ella)
Associate Editor, Communications Earth & Environment
Consulting Editor, Communications Sustainability

EDITORIAL POLICIES AND FORMATTING

Editorial Policy: [Policy requirements](https://www.nature.com/documents/nr-editorial-policy-checklist.pdf) (Download the link to your computer as a PDF.)

- Behavioural and social science
- Ecological, evolutionary & environmental sciences
- Life sciences

<https://www.nature.com/documents/nr-reporting-summary.zip>

Furthermore, please align your manuscript with our format requirements, which are summarized on the following checklist: [Communications Earth & Environment formatting checklist](https://www.nature.com/documents/commsj-phys-style-formatting-checklist-article.pdf)

and also in our style and formatting guide [Communications Earth & Environment formatting guide](https://www.nature.com/documents/commsj-phys-style-formatting-guide-accept.pdf) .

*** DATA: Communications Earth & Environment endorses the principles of the Enabling FAIR data project (<http://www.copdess.org/enabling-fair-data-project/>). We ask authors to make the data that support their conclusions available in permanent, publically accessible data repositories. (Please contact the editor if you are unable to make your data available).

All Communications Earth & Environment manuscripts must include a section titled "Data Availability" at the end of the Methods section or main text (if no Methods). More information on this policy, is available at <http://www.nature.com/authors/policies/data/data-availability-statements-data-citations.pdf>.

If a community resource is unavailable, data can be submitted to generalist repositories such as [figshare](https://figshare.com/) or [Dryad Digital Repository](http://datadryad.org/). Please provide a unique identifier for the data (for example a DOI or a permanent URL) in the data availability statement, if possible. If the repository does not provide identifiers, we encourage authors to supply the search terms that will return the data. For data that have been obtained from publically available sources, please provide a URL and the specific data product name in the data availability statement. Data with a DOI should be further cited in the methods reference section.

REVIEWER COMMENTS:

Reviewer #1 (Remarks to the Author):

This study investigates ocean acidification during the Toarcian Oceanic Anoxic Event (T-OAE, ~183 Ma), a period of global climate change and marine extinctions likely driven by CO₂ and CH₄ emissions from Karoo-Ferrar volcanism. The authors analyze boron isotopes (δ¹¹B) from micrite, brachiopods, and bivalves in marine sections from Spain and Portugal to reconstruct ocean pH changes. Their results show that only micrite records a temporary decline in δ¹¹B during the T-OAE, supporting the hypothesis of ocean acidification, while brachiopods and bivalves exhibit stable δ¹¹B values, suggesting they may have buffered changes in ocean pH. Using a coupled biogeochemical model (the classical COPSE model), they link the observed pH changes to carbon cycle disturbances, including volcanic CO₂ release and methane degassing, and compare these findings to faunal turnover patterns during the event.

I find this study interesting in refining how ocean acidification events are reconstructed in the geologic record using micrite. It highlights the limitations of using brachiopods and bivalves as pH proxies and argues that micrite might better capture transient acidification signals. The integration of multi-isotopic data with biogeochemical modeling seems to support ocean acidification during the T-OAE. If correct, this work improves our understanding of past carbon cycle perturbations and provides insights relevant to modern climate change scenarios.

I am generally supportive of the publication of this study but have two major concerns that need to be addressed—one related to the boron isotope records and the other to the COPSE model. While I am not certain whether these issues will substantially alter the paper's conclusions, they are critical enough that they should be fully addressed.

1. My primary concern regarding the boron isotope records is the use of micrite as a proxy for ocean pH. In the main text, the authors state that "Modern micrite derives primarily from disaggregated, diverse skeletal carbonates and is a variable mixture of aragonite, high-Mg calcite, and low-Mg calcite formed by the recrystallization of lime mud. It is, however, unsuitable for calibration, so accurate δ¹¹B-ocean pH calibration and hence ocean pH assessment is not possible." Similarly, in the supplementary materials, they reiterate that "An accurate δ¹¹B-ocean pH calibration using micritic carbonates is not possible. The composition of Pliensbachian–Toarcian micrite is unknown and could be of biological and non-biological origin, material for which we do not have a verified calibration."

There are very few studies that have attempted to calibrate boron isotope-based pH reconstructions using modern micrite. One of those studies is Zhang et al. (2017, EPSL), which found that micrite-based pH reconstructions showed large offsets from in situ pH values, likely due to microenvironmental effects influencing the pH signal during micrite formation. Given this, I think the authors should provide a more in-depth discussion on why micrite may not reliably record seawater pH and what factors could influence its δ¹¹B signature. This is particularly important because it affects how the authors interpret the δ¹¹B record from micrite during the Toarcian Oceanic Anoxic Event. While the authors argue that "reproducing identical patterns in two disparate sections across the same time interval argues for an original and inter-regional record," this pattern does not necessarily confirm that the δ¹¹B signal reflects ocean acidification. Alternative explanations could be that deeper-dwelling carbonate-secreting organisms migrated to shallower waters in both sections during the perturbation event, recording lower δ¹¹B values, or that the composition of organisms contributing to micrite formation shifted similarly in both locations, producing a parallel δ¹¹B trend. Without a clearer understanding of micrite composition evolution and its response to seawater pH changes, it is difficult to draw firm conclusions about using the micrite δ¹¹B signal to track ocean acidification during the T-OAE.

2. My second concern is related to how the COPSE model calculates alkalinity. The model tracks the total carbon reservoir in the ocean-atmosphere system and then applies an atmospheric fractionation factor to estimate atmospheric pCO₂ and oceanic dissolved inorganic carbon (DIC). From what I see, the authors seem to use a similar method to calculate alkalinity, but I believe this requires more explanation and justification. While DIC and alkalinity are closely related, they are not always tightly coupled—certain processes can affect one without directly influencing the other. Given that ocean pH calculations (and their comparison to δ¹¹B records) depend on robust estimates of both pCO₂ (or DIC) and alkalinity, I think the authors need to clarify their approach to calculating alkalinity and justify why it is valid for this study. A more detailed discussion on this point would strengthen confidence in the model's ability to accurately reconstruct pH changes through the T-OAE.

Below is my minor comment:

Line 187-188: To calculate pH, DIC or alkalinity (coupled with pCO₂) are also needed. Please clarify or correct.

Reviewer #2 (Remarks to the Author):

The manuscript by Kasemann and co-authors presents an interesting and valuable dataset from two well-studied sections in Spain and Portugal, aiming to constrain ocean acidification during the Toarcian Oceanic Anoxic Event (T-OAE). The authors suggest that the lack of δ¹¹B variation in shell fossils, alongside the observed δ¹¹B variations in micrite from both sections, supports the hypothesis of ocean acidification during the T-OAE, with micrite being a more reliable proxy for ancient ocean pH.

While this is an intriguing hypothesis, I believe there are two key aspects that warrant further consideration and clarification:

1. The assumption of ocean acidification during the T-OAE: The authors' argument hinges on the assumption that ocean acidification occurred during this event. However, the $\delta^{11}\text{B}$ data from brachiopods and bivalves do not seem to support this assumption. It would be helpful to further explore and discuss the potential implications of the $\delta^{11}\text{B}$ data from these fossils, as they may suggest that ocean acidification did not occur as expected. The authors may wish to address this discrepancy and consider alternative explanations for the $\delta^{11}\text{B}$ trends observed in their dataset.

2. The timing of the $\delta^{11}\text{B}$ excursion in the two sections: The exact timing of the $\delta^{11}\text{B}$ excursion differs between the Portuguese and Spanish sections. In the Portugal section, the lowest pH is observed when the carbon isotope values are most negative, while in Spain, the lowest pH occurs toward the end of the T-OAE. This difference is difficult to reconcile with simple changes in sedimentary rates or cryptic hiatuses, and further explanation is needed. The authors could explore this timing discrepancy in more detail and consider whether other factors, such as basin restriction or local depositional environment, might account for these variations.

Additionally, there is a concern regarding the potential diagenetic overprint of the $\delta^{18}\text{O}$ data from micrite in Portugal, which may affect the interpretation of the $\delta^{11}\text{B}$ signal. The authors should consider how susceptible the $\delta^{11}\text{B}$ signal is to diagenetic alteration, particularly since the micrite samples are also subject to such processes. A discussion on the extent of diagenetic overprint and its potential impact on the $\delta^{11}\text{B}$ data would strengthen the manuscript.

Given these points, I recommend that the manuscript undergo major revisions to address these concerns before further consideration. I believe that with additional clarification and discussion of these issues, the manuscript could make an important contribution to the understanding of ocean pH and acidification during the T-OAE.

Below are some detailed comments and suggestions:

Line 32: delete "be".

Line 48: change "negative large-scale" to "large-magnitude negative".

Line 51: change "rising limb" to "recovery".

Line 52-54: what about CO₂ removal by enhanced organic carbon burial?

Line 110: removal "the same"

Lines 138-140: I don't see any decrease in $\delta^{18}\text{O}$ values from micrite for the Portugal record, but rather a slight increase. What does this mean in terms of diagenetic overprint on $\delta^{18}\text{O}$ as well as $\delta^{11}\text{B}$ values?

Lines 144-147: due to the data gap in most of the T-OAE, an alternative way to interpret the $\delta^{11}\text{B}$ data of bivalves (Spain) is that the data in general fits with the values and trends generated from micrite.

Lines 150-153: following the above comment, the $\delta^{18}\text{O}$ micrite values are lighter, but also different trends than the $\delta^{18}\text{O}$ fossil values. Could this indicate diagenetic overprint on the sample?

Line 160: for "subsequent", do you mean "stratigraphically above"?

Line 165: change "middle and early Toarcian" to "early and middle Toarcian". What does the word "respectively" refer to?

Line 166: change "before" to "prior to".

Lines 206-208: what is the role of organic carbon burial in bringing the system back here? It has been suggested to be an important carbon sink for this event, especially with the widespread organic-rich black shale deposition at this time.

Line 277: delete "is".

Line 278: how does differences in sedimentation rate influence the relative timing of pH change to the T-OAE?

Lines 297-299: The $\delta^{11}\text{B}$ data shows a recovery to pre-Toarcian conditions during the Serpentinum zone, instead of at the end.

Lines 331-332: depending on how you look at the data, it is also possible to say that the minimum pH coincides with the abrupt increase in shell size.

Supplementary materials:

Age model: why the Geological Time Scale 2016 is used, instead of the more recent ones.

Line 217: should it be "cannot" or "can" account for the offset?

Reviewer #3 (Remarks to the Author):

Review of Kasemann et al.

The manuscript by Kasemann et al. makes the case that the B-isotope composition of micrite provides a reliable and robust indicator of ocean pH changes in deep time, and that the data obtained across the T-OAE reveal a signal of acidification coincident with carbon release and warming. The work is very well-written and the illustrations are good. The manuscript is well-structured and well-referenced. Overall, the manuscript provides important and novel data and I recommend publication after minor revision.

Main comments

I am not an expert in B-isotope systematics, so I cannot find fault with the technical details of the work. I have only minor comments on the modeling approach and some of the text relating to the way the T-OAE is presented.

One issue, which is admittedly minor in the context of the work, is that I do think we need to move away from lumping the Karoo and Ferrar LIPs into a single entity (e.g., lines 41, 285 etc.). These LIPs were distinct in onset timing and genesis, and recent work has indicated that it was Ferrar that was coeval with the T-OAE, and Karoo started earlier and likely initiated climate changes at the PI-To. The literature citing Karoo as the event coincident with the T-OAE did so because of the incorrect assumption that the T-OAE was older than we actually now know it was (see Al-Suwaidi et al., 2022; Kemp et al., 2024).

I'm not a huge fan of the approach the authors use for source modeling. In all experiments, a biogenic CH₄ component is assumed (–60‰) along with a volcanic CO₂ component. I think it would have been more sensible to avoid assuming a mix of 2 specific C sources and instead derive a single average value for the likely isotopic composition of the total C. From that, the likely contribution of different sources (biogenic, volcanic etc.) could be discussed later. At present, the results and figures show how a specific balance of carbon from hydrate CH₄ (–60‰) and volcanic CO₂ could fit the data. But couldn't the same results be achieved if the carbon was a different mix of volcanic CO₂ and thermogenic CO₂/CH₄ (i.e. from sill intrusion into organic-rich rocks, ~–30‰)?

Part of the reason I ask is because the potentially important role of thermogenic C released during sill intrusion is effectively ignored (see for example: McElwain et al., 2005; Svensen et al., 2007, 2012; Heimdal et al., 2021; Kemp et al., 2024). Average C source compositions of –11‰ and –15‰ are implied by cGENIE modelling for the PETM and Permian-Triassic, and these values are perhaps most likely to have been achieved by a mix of volcanic and thermogenic sources – without the need for any/much biogenic CH₄ (Gutjahr et al., 2017; Cui et al., 2021). This is not to say methane was not involved during the T-OAE (the astronomical pacing of the CIE supports it), but it just seems like an oversight to ignore a thermogenic contribution. Maybe too late now to change anything since the models have been run, but my suggestion is that the authors maybe note the assumptions being made about the mix of two specific sources, and add a line or two to note that the carbon could have been a mix of volcanic and thermogenic (if the authors agree that this is reasonable).

Line by line comments

Line 43. "as high as". Both Ruebsam et al. (2020) and McElwain et al. (2005) have data to suggest it was higher than 900 ppm.

Line 51. Might be good to consider alternative wording here. "rising limb..." may not mean much to the reader, and maybe better to define it in terms of a stabilization of values or the end of a positive shift etc.

Line 53. Not that 'recent' now. Even more recently, a CIE duration of ~300 kyr (and almost certainly <407 Kyr) has been constrained by CA-ID-TIMS geochronology (Kemp et al., 2024). Too late now for the modelling, but would be interesting to know whether this dramatically effects things. Maybe a line could be added to briefly note the implications of a faster release of carbon than the manuscript currently assumes?

Line 110: What makes them 'well-preserved' successions?

Line 131: "We place all data into the same biostratigraphic...". Repetition of something already clear in the last section.

Line 145, 161, 231: 'data gaps'. Because of the lack of suitable material, lack of any material, lack of sampling, or lack of good outcrop? I know Methods is cited here but a single line in the main text to explain why we have these gaps could be helpful.

Al-Suwaidi, A.H. et al., 2022. New age constraints on the Lower Jurassic Pliensbachian–Toarcian Boundary at Chacay Melehue (Neuquén Basin, Argentina). *Scientific Reports*, 12, 4975.
Cui, Y. et al., 2021. Massive and rapid predominantly volcanic CO₂ emission during the end-Permian mass extinction. *Proc. Natl. Acad. Sci. U.S.A.*, 118, e2014701118.
Gutjahr, M. et al., 2017. Very large release of mostly volcanic carbon during the Palaeocene–Eocene Thermal Maximum. *Nature*, 548, 573–577.
Heimdal, T.H., 2021. Assessing the importance of thermogenic degassing from the Karoo Large Igneous Province (LIP) in driving Toarcian carbon cycle perturbations. *Nature Communications*, 12, 6221.
Kemp, D.B. et al., 2024. The timing and duration of large-scale carbon release in the Early Jurassic. *Geology*, doi:10.1130/G52457.1.
McElwain, J.C. et al., 2005. Changes in carbon dioxide during an oceanic anoxic event linked to intrusion into Gondwana coals. *Nature*, 435, 479–495.
Ruebsam, W. et al., 2020. $\delta^{13}\text{C}$ of terrestrial vegetation records Toarcian CO₂ and climate gradients. *Scientific Reports*, 10, 117.
Svensen, H. et al., 2012. Rapid magma emplacement in the Karoo Large Igneous Province: Earth and Planetary Science Letters. 325–326, 1–9.
Svensen, H. et al., 2007. Hydrothermal venting of greenhouse gases triggering Early Jurassic global warming. *Earth Planetary Science Letters*, 256, 554–566.

Communications Earth & Environment is committed to improving transparency in authorship. As part of our efforts in this

direction, we are now requesting that all authors identified as 'corresponding author' create and link their Open Researcher and Contributor Identifier (ORCID) with their account on the Manuscript Tracking System prior to acceptance. ORCID helps the scientific community achieve unambiguous attribution of all scholarly contributions. You can create and link your ORCID from the home page of the Manuscript Tracking System by clicking on 'Modify my Springer Nature account' and following the instructions in the link below. Please also inform all co-authors that they can add their ORCIDs to their accounts and that they must do so prior to acceptance.

Version 1:

Decision Letter:

Dear Professor Kasemann,

Your manuscript titled "Ocean acidification at the Toarcian Anoxic Event captured by boron isotopes in the lime mud record" has now been seen by our reviewers, whose comments appear below. In light of their advice we are delighted to say that we are happy, in principle, to publish a suitably revised version in Communications Earth & Environment.

We therefore invite you to revise your paper one last time to address the remaining concerns of our reviewers. At the same time we ask that you edit your manuscript to comply with our format requirements and to maximise the accessibility and therefore the impact of your work.

EDITORIAL REQUESTS:

*****Please take care to match our formatting and policy requirements. We will check revised manuscript and return manuscripts that do not comply. Such requests will lead to delays.*****

SUBMISSION INFORMATION:

OPEN ACCESS:

Communications Earth & Environment is a fully open access journal. Articles are made freely accessible on publication. For further information about article processing charges, open access funding, and advice and support from Nature Research, please visit <https://www.nature.com/commsenv/open-access>

Link Redacted

Best regards,

Alice Drinkwater, PhD
Associate Editor
Communications Earth & Environment
Consulting Editor
Communications Sustainability

REVIEWERS' COMMENTS:

Reviewer #1 (Remarks to the Author):

The authors have provided a thoughtful and thorough response to the initial critiques. I find their justification convincing and have no further comments.

Reviewer #2 (Remarks to the Author):

I appreciate the authors' efforts in thoroughly addressing the reviewers' previous comments. The manuscript has significantly improved as a result. The authors have clarified that the boron isotope data from micrite and carbonate microfossils are used to support the modelled results, and they now provide a more detailed discussion of vital effects in brachiopod and bivalve shells—specifically explaining why these organisms may not record pH changes as expected. However, it would strengthen the argument to further elaborate on why micrite is considered a reliable archive of pH. As noted in the introduction, micrite is thought to originate primarily from algae, microbes, or more likely from calcareous nannoplankton (e.g., *Schizosphaerella*, coccoliths, dinoflagellates), and may also include admixtures of peri-platform ooze. It would be helpful if the authors could provide supporting evidence or references demonstrating that these sources are capable of recording seawater pH without significant influence from biological vital effects. I recommend publication pending minor revisions to address this point.

Minor suggestions (line numbers from the tracked changes version)

Line 22: it would be good to spell out Ma in first use.

Line 33: what does interrogated taxa mean? It also does not read well to have interrogated and interrogation in the same sentence.

Line 50: change "found" to "observed"

Lines 50-51: change to "observed in fossil wood, diverse marine bulk organic and inorganic substrates, and carbonate microfossils"

Lines 131: add "soly" before "using the B isotope records"

Line 184: delete "did not"

Line 301: change "extinctions events" to "extinction events"

Reviewer #3 (Remarks to the Author):

I am satisfied that the reviewers have addressed my concerns and queries and I have no further comments

RE: “Ocean acidification at the Toarcian Anoxic Event captured by boron isotopes in the lime mud record” by Kasemann et al.

We would like to thank the reviewers for three insightful reviews with constructive comments and positive endorsements.

To facilitate the evaluation of the changes made, we have (i) copied the comments/suggestions of all three reviewers in blue/non-italics and (ii) provided the response to each comment/suggestion in black/italics. All line numbers cited in our responses refer to (1) the revised track changes and (2) the revised clean version of the manuscript (e.g., line 1/2). In the files with the tracked changes of the revised version of the manuscript and the supplement, the text marked in red is new or changed, the text marked in green has only been moved.

Reviewer #1 (Remarks to the Author):

This study investigates ocean acidification during the Toarcian Oceanic Anoxic Event (T-OAE, ~183 Ma), a period of global climate change and marine extinctions likely driven by CO₂ and CH₄ emissions from Karoo-Ferrar volcanism. The authors analyze boron isotopes (δ¹¹B) from micrite, brachiopods, and bivalves in marine sections from Spain and Portugal to reconstruct ocean pH changes. Their results show that only micrite records a temporary decline in δ¹¹B during the T-OAE, supporting the hypothesis of ocean acidification, while brachiopods and bivalves exhibit stable δ¹¹B values, suggesting they may have buffered changes in ocean pH. Using a coupled biogeochemical model (the classical COPSE model), they link the observed pH changes to carbon cycle disturbances, including volcanic CO₂ release and methane degassing, and compare these findings to faunal turnover patterns during the event.

I find this study interesting in refining how ocean acidification events are reconstructed in the geologic record using micrite. It highlights the limitations of using brachiopods and bivalves as pH proxies and argues that micrite might better capture transient acidification signals. The integration of multi-isotopic data with biogeochemical modeling seems to support ocean acidification during the T-OAE. If correct, this work improves our understanding of past carbon cycle perturbations and provides insights relevant to modern climate change scenarios.

I am generally supportive of the publication of this study but have two major concerns that need to be addressed—one related to the boron isotope records and the other to the COPSE model. While I am not certain whether these issues will substantially alter the paper’s conclusions, they are critical enough that they should be fully addressed.

1. My primary concern regarding the boron isotope records is the use of micrite as a proxy for ocean pH. In the main text, the authors state that “Modern micrite derives primarily from disaggregated, diverse skeletal carbonates and is a variable mixture of aragonite, high-Mg calcite, and low-Mg calcite formed by the recrystallization of lime mud. It is, however, unsuitable for calibration, so accurate δ¹¹B-ocean pH calibration and hence ocean pH assessment is not possible.” Similarly, in the supplementary materials, they reiterate that “An accurate δ¹¹B-ocean pH calibration using micritic carbonates is not possible. The composition of Pliensbachian–Toarcian micrite is unknown and could be of biological and non-biological origin, material for which we do not have a verified calibration.”

There are very few studies that have attempted to calibrate boron isotope-based pH reconstructions using modern micrite. One of those studies is Zhang et al. (2017, EPSL), which found that micrite-

based pH reconstructions showed large offsets from in situ pH values, likely due to microenvironmental effects influencing the pH signal during micrite formation. Given this, I think the authors should provide a more in-depth discussion on why micrite may not reliably record seawater pH and what factors could influence its $\delta^{11}\text{B}$ signature. This is particularly important because it affects how the authors interpret the $\delta^{11}\text{B}$ record from micrite during the Toarcian Oceanic Anoxic Event. While the authors argue that “reproducing identical patterns in two disparate sections across the same time interval argues for an original and inter-regional record,” this pattern does not necessarily confirm that the $\delta^{11}\text{B}$ signal reflects ocean acidification. Alternative explanations could be that deeper-dwelling carbonate-secreting organisms migrated to shallower waters in both sections during the perturbation event, recording lower $\delta^{11}\text{B}$ values, or that the composition of organisms contributing to micrite formation shifted similarly in both locations, producing a parallel $\delta^{11}\text{B}$ trend. Without a clearer understanding of micrite composition evolution and its response to seawater pH changes, it is difficult to draw firm conclusions about using the micrite $\delta^{11}\text{B}$ signal to track ocean acidification during the T-OAE.

(i) It is true that there are very few studies dealing with boron isotope-based pH reconstructions using modern micrites or rather whole rock carbonates, and the studies we know of are all focused on the Great Bahama Bank.

The most detailed study that has attempted a possible calibration is indeed the study by Zhang et al. (2017, EPSL). This work nicely demonstrates how variable the B isotope composition of carbonates accumulated on a modern marine reef platform can be, and that care must be taken when using such material as proxy archives. The material analyzed by Zhang et al. (2017) was a variable mixture of aragonite, high-Mg calcite and low-Mg calcite mainly accumulated on the platform near sea level. At least some of the lime mud derived from altered biogenic carbonates and some of the investigated material was exposed during low tide, deposited in a salt pond and represents reworked reef cements from a reef slope cavity. Considering that different types of carbonates can have different B concentrations and more importantly, different fractionation factors for B isotopes, it is not surprising, and indeed to be expected, that such a heterogeneous mixture will show an unsystematic scatter without uniformity in B concentration and isotope composition and therefore be unsuitable for calibrating ocean pH based on boron isotopes.

Two other studies deal with core material from the toe-of-slope of the western Bahama Bank. Both cores, Site 1007 (Zhao et al. 2023, American Journal of Science) and Clino Core (Stewart et al. 2015, Geology), are located close to each other and show a comparable Pleistocene to late Pliocene material composition or lithology. This changes from an aragonite-rich upper section (needles and peloids) to a lower calcite-rich section dominated by skeletal debris, including benthic/planktic foraminifera, corals, red/green algae, echinoderms, molluscs, calcareous nannofossils, fish debris. Also, unidentifiable blackened grains with turbidites and slumped intervals and potentially redeposited carbonates from the platform formed during seawater lowstands (information from Shipboard Scientific Party reports, 1997). For the Clino core, the samples cover the entire zone of meteoric diagenesis and merge into the mixing diagenetic zone and were also used by Stewart et al. (2015) to explore the influence of meteoric diagenesis on the boron system. In terms of a suitable calibration, the question here is how much of the changing B isotope composition and B concentration is diagenetic and how much of it is natural variation caused by the skeletal debris.

In contrast to the material sampled from the Bahama bank, our carbonates are not a comparable heterogeneous mixture, they were not exposed during lowstands and are rigorously checked for meteoric influence. As detailed in the method section in “Sample preparation - micritic material” from line 555 in the track changes version and line 509 in the cleaned version, we tried to sample only the homogeneous micrite matrix and avoided any obvious fossil debris and areas of burrow traces and

veins. We also added a new chapter on “Potential influence of contamination and diagenesis on micrite” (from line 736 in track changes/624 cleaned version) in the methods section summarizing all optical and geochemical tests performed on the micrite to check for indications of contamination, alteration and meteoric diagenesis that could have influenced our samples and results (as suggested by reviewer 2).

We speculate in the main text (line 105/98) that algae, microbes or most likely calcareous nannoplankton are the main sources of the sampled micrite and provided several references (line 107/100) that calcareous nannoplankton, coccoliths and dinoflagellates are abundant and may have produced the majority of the micrite analysed. This collection of material is of course also quite heterogeneous to some extent, but on a much smaller spatial scale and seems to homogenise the collected environmental signal. This may seem contradictory at first, but in principle it is exactly what we do when we use foraminifera or corals as archives. We know from in-situ techniques such as laser and ion probe that the B isotope composition in a foraminifera test (from chamber to chamber) and in a coral (sometimes in micrometre scale) is usually highly variable. But once we have a representative sample that we analyse wet-chemically with TIMS or the MC-ICP-MS, we homogenise the values and can use the skeletal carbonate material as an archive for B isotope-ocean pH calibrations. The same also applies to other isotope systems such as carbon and oxygen isotopes. That we do not collect micrite with strongly fluctuating composition is also supported by the uniformly low B concentrations and the reproduced smooth change in the B isotope pattern (as explained below).

Just as the Bahama Bank material chosen has proven unsuitable for calibration, we are not aware of any other carbonate production area that would be suitable for boron isotope pH calibration of ancient micrites based on modern ones, and I guess we regret this fact the most, as a robust calibration is at the moment not possible. But we believe that we can track relative changes.

(ii) We also agree with the reviewer that the “reproducing identical patterns in two disparate sections across the same time interval” does not per se confirm that the $\delta^{11}\text{B}$ signal reflects ocean acidification. Therefore, we explored other factors such as changes in temperature, salinity, bathymetry, weathering and source material that may influence the B isotope composition of carbonates even when seawater pH was stable and added a note (from line 360/320) with referral to the supplementary discussion „Environmental controls on the $\delta^{11}\text{B}$ composition of micrite”.

In the supplementary discussion under “Environmental controls on the $\delta^{11}\text{B}$ composition of micrite” we discussed in detail that the calculated rise in seawater temperature at the site of carbonate precipitation ($\sim 3.5 \pm 0.3$ °C) could cause a change in marine carbonate B isotope composition, but this would be an increase rather than a decrease as observed. Similar to temperature, salinity can also change the B isotope composition in carbonates. However, to cause such a temporary decrease in $\delta^{11}\text{B}$ values as observed in the micrite, a drastic decrease in salinity (e.g., from 35 psu to 25 psu) is required, for which there is no evidence. Both of our study sites are from the mid- to outer-ramp environment below the storm-wave base, far from direct freshwater inputs.

In addition, changing water depth can also affect the B isotope composition in carbonates, but a potential depth effect is considered minor for both sections as the depositional environment is relatively stable and is likely to have remained below the storm-wave base throughout the studied time interval. Enhanced weathering was also discussed as an influencing factor, but this would lead to increased ocean alkalinity and increased buffering capacity of the seawater in the potentially restricted basin. An increased boron influx caused by enhanced continental weathering could have also led to a temporary decrease in the B isotopic composition of seawater and thus to a decrease in the B isotopic composition in carbonates. However, given the residence time of the boron and the

timing of the excursion, this is unlikely even in a potentially confined basin and would inevitably have had an impact on the B isotope composition of brachiopods and oysters.

In the interest of completeness, we have now included this additional point regarding weathering in the discussion (line 363/320) and referred to the detailed information and explanation in the supplementary discussion under “Environmental controls on the $\delta^{11}\text{B}$ composition of micrite”. We initially considered moving the section “Environmental controls on the $\delta^{11}\text{B}$ composition of micrite” from the supplement to the methods section of the main text to make the discussion of alternative scenarios and potential constraints more accessible to the reader, but also to demonstrate that we find no evidence that environmental processes other than acidification could have caused the temporary decrease in $\delta^{11}\text{B}$ values (from line 360/317). However, we retained the section in the supplement since testing whether environmental processes other than ocean acidification control the $\delta^{11}\text{B}$ composition of carbonates should be a prerequisite in evaluating the data anyhow, and not just for micrite.

As alternative scenarios leading to a parallel $\delta^{11}\text{B}$ trend, the reviewer suggested that either deeper-dwelling, carbonate-secreting organisms migrate to shallower waters during the perturbation event in both sections and record lower $\delta^{11}\text{B}$ values, or that the composition of organisms contributing to micrite formation shifted similarly in both locations. This is indeed worth considering, and to some extent we have already done so in the supplementary discussion, and we refer to this in the manuscript (from line 104/97 and line 360/317).

As initially outlined in the supplementary information under “Calculation of seawater pH using boron isotope composition in lime mud (micrite)” the skeletal material of macroinvertebrates in the Pliensbachian and Toarcian most likely derived from brachiopods, bivalves, gastropods and ammonites, and possible sources of pelagic calcifiers are coccoliths. The benthic skeletal material usually has significantly higher boron mass fraction than the micrite so that a shift in the B isotope composition should be accompanied by an increase in the B concentration, but this is not observed. We have moved this paragraph to our considerations on “Environmental controls on the $\delta^{11}\text{B}$ composition of micrite” – Source Material.

The T-OAE is of course associated with a faunal loss and turnover, whereby the opportunistic brachiopod species *Soaresirhynchia bouchardi* dominates the T-OAE (line 296/285, 434/381). Although this species has comparatively low boron mass fractions, it has a significantly higher boron isotope composition than the micrite (Supplementary Discussion). When we look at pelagic biota, foraminifera come to mind. However, planktonic foraminifera only appeared in the late Early Jurassic, at the earliest in the mid-Toarcian Bifrons zone, i.e., after the T-OAE and the B isotope excursion. Benthic foraminifera could of course also be considered, but from what we know, benthic foraminifera can have low B amount contents at depth, similar to the micrite, but they usually have much higher contents of up to 25 $\mu\text{g/g}$ (Rae et al. 2011, EPSL) or even 65 $\mu\text{g/g}$ (Levi et al. 2019, Front. Earth Sci.). As such, a temporary contribution of benthic foraminifera should also be obvious via a change in the B amount content. We nevertheless add this consideration in the Supplementary Discussion on Source Material.

We were also very careful in our sample selection and preparation, picked only the micrite matrix while avoiding any shells of macroinvertebrates and areas of burrow traces and veins, as explained in the method section from line 555/509. Nevertheless, we cannot completely rule out a transport of deeper dwelling, carbonate-secreting organisms that migrated to shallower waters during the perturbation, although we don't have any evidence for this.

As mentioned above, it is indicated in the main text (line 105/98) that it is more likely that algae, microbes or most likely calcareous nannoplankton are the main sources of micrite. We have provided several references (line 107/100) that calcareous nannoplankton such as *Schizosphaerella*, coccoliths

and dinoflagellates are abundant and may have produced the majority of the micrite sampled. The micrite could also represent an admixture of peri-platform ooze from a proximal carbonate platform, potentially supplying aragonitic fines.

As we are quite aware of the problematic nature of using micrite as an archive for $\delta^{11}\text{B}$ -ocean pH, in this study we took a different approach to test ocean acidification and the ability of the micrite to track changes in ocean pH. Normally, one produces a B isotope record from carbonate skeletal material (e.g., forams, corals, micrite) and calculates an ocean pH record. We have also done this for the Neoproterozoic and for the PTB using micrite. In this study, we started by reconstructing a record of ocean pH, carbon cycle disturbance and rapidly changing environmental conditions using the COPSE model and previously published data e.g., carbon isotopes. We have tried to make this more explicit in the revised text (from line 131/123).

We then compared the modelled pH record with our B isotope data from the different carbonate components (micrite, brachiopods and oyster shells). The pattern of the modelled ocean pH was found to be broadly consistent with the micrite B isotope patterns, supporting our hypotheses that ocean pH is the major driver for the observed $\delta^{11}\text{B}$ changes in the micrite. In contrast, the fossil record shows no comparison and as such now indication for changes in ocean pH. Finally, we compared our boron proxy record of ocean pH to palaeoecological patterns established for the Spanish and Portuguese sections. This allowed us to conclude, that micrite derived $\delta^{11}\text{B}$ values can capture trends in ocean pH changes.

2. My second concern is related to how the COPSE model calculates alkalinity. The model tracks the total carbon reservoir in the ocean-atmosphere system and then applies an atmospheric fractionation factor to estimate atmospheric pCO₂ and oceanic dissolved inorganic carbon (DIC). From what I see, the authors seem to use a similar method to calculate alkalinity, but I believe this requires more explanation and justification. While DIC and alkalinity are closely related, they are not always tightly coupled—certain processes can affect one without directly influencing the other. Given that ocean pH calculations (and their comparison to d11B records) depend on robust estimates of both pCO₂ (or DIC) and alkalinity, I think the authors need to clarify their approach to calculating alkalinity and justify why it is valid for this study. A more detailed discussion on this point would strengthen confidence in the model's ability to accurately reconstruct pH changes through the T-OAE.

We agree with the reviewer regarding the decoupling between total alkalinity and DIC. Please see Supplementary Model description, equations (2) and (3), which illustrate how DIC and total alkalinity are explicitly resolved separately from each other at each timestep. As the reviewer correctly states, DIC is essentially the ocean fraction of the global ocean-atmosphere CO₂ reservoir (pCO₂ being calculated by the atmospheric fraction). Carbonate alkalinity scales linearly with DIC. Total alkalinity is the weighted sum of carbonate alkalinity and Borate alkalinity:

$$[TAlk]_{(t)} = TAlk_0 \cdot \left(\frac{CAlk_0}{TAlk_0} \cdot \frac{A(t)}{A_0} \cdot (1 - \phi(t)) + \frac{B(t)}{B_0} \cdot \frac{BAlk_0}{TAlk_0} \right)$$

Total alkalinity, pCO₂, temperature, and surface ocean phosphate concentration, each of which are estimated by the box model, are fed into CO2SYS in order to calculate the pH estimate (which is then used to reconstruct the $\delta^{11}\text{B}$ proxies as described). Thus, the estimate of total alkalinity already incorporates the effect of borate and phosphate, two of the main factors by which it is likely to differ from carbonate alkalinity.

Below is my minor comment:

Line 187-188: To calculate pH, DIC or alkalinity (coupled with pCO₂) are also needed. Please clarify or correct. We added the information in line 212/202. pCO₂, total alkalinity (carbonate plus borate), temperature and surface phosphate concentration are all technically inputs for the pH estimate. Borate and phosphate are two components of alkalinity that can change “on the timescales under consideration”. There are several other components of alkalinity but they have such long residence times they can’t vary on our timescale of interest.

Reviewer #2 (Remarks to the Author):

The manuscript by Kasemann and co-authors presents an interesting and valuable dataset from two well-studied sections in Spain and Portugal, aiming to constrain ocean acidification during the Toarcian Oceanic Anoxic Event (T-OAE). The authors suggest that the lack of $\delta^{11}\text{B}$ variation in shell fossils, alongside the observed $\delta^{11}\text{B}$ variations in micrite from both sections, supports the hypothesis of ocean acidification during the T-OAE, with micrite being a more reliable proxy for ancient ocean pH. While this is an intriguing hypothesis, I believe there are two key aspects that warrant further consideration and clarification:

1. The assumption of ocean acidification during the T-OAE: The authors' argument hinges on the assumption that ocean acidification occurred during this event. However, the $\delta^{11}\text{B}$ data from brachiopods and bivalves do not seem to support this assumption. It would be helpful to further explore and discuss the potential implications of the $\delta^{11}\text{B}$ data from these fossils, as they may suggest that ocean acidification did not occur as expected. The authors may wish to address this discrepancy and consider alternative explanations for the $\delta^{11}\text{B}$ trends observed in their dataset.

We agree with the reviewer that one has to always consider alternative explanations for the $\delta^{11}\text{B}$ trends observed in the dataset, and we have done just that in the original submission in the supplementary information in “Environmental controls on the $\delta^{11}\text{B}$ composition” and “Calculation of seawater pH using boron isotope composition in lime mud (micrite)”. In these two sections, we explored other mechanisms such as changes in temperature, salinity, bathymetry, weathering and source material that is able to influence the B isotope composition of carbonates, even when ocean pH was stable. We concluded that while all of these processes may have an influence on B isotope composition, none of them could have been the main cause of the observed transient decrease in the record. We addressed this in more detail in our response to Reviewer #1. However, since both Reviewer #1 and Reviewer #2 emphasised this, we realised that our discussion on this topic was not obvious enough, we pointed our considerations regarding alternative explanations for the $\delta^{11}\text{B}$ trends such as temperature, salinity, bathymetry, weathering and micritic source material out and inserted a sentence in the manuscript (line 362 in track changes and line 318 in cleaned version) to draw the reader's attention to the complications and our considerations detailed in the supplementary discussion now summarized under “Environmental controls on the $\delta^{11}\text{B}$ composition of micrite”.

While our model output and the B isotope record of the micrite support an ocean acidification event, the records of the brachiopods and bivalves do not. If we accept that we are observing an ocean acidification event, this implies that the brachiopods and bivalves studied were able to buffer changes

in ocean pH. In the original submission, we briefly mentioned the possibility of the buffering capacity of the brachiopods and bivalves in the main text and left the main argumentation for the supplement. Since we agree with the reviewer that this discrepancy deserves to be more fully addressed, we have moved parts of the text from the supplement to the discussion in the main text and included further support for our assumption that brachiopods (from line 273/261) and bivalves (from line 312/294) do not necessarily record seawater pH in their shells.

In summary, we refer to studies that have addressed the potential use of brachiopod $\delta^{11}\text{B}$ values as pH tracers. They have shown that the calcite of some modern terebratulid brachiopod shells records changes in $\delta^{11}\text{B}$ values with changing ocean pH conditions. However, it is also clear from all three studies that the pH of the internal calcifying fluids is significantly regulated on a species-specific basis, so that the use of brachiopod $\delta^{11}\text{B}$ values to reconstruct changes in ocean pH conditions is restricted to single species or by cross-calibrating coeval species, and in principle only to some modern terebratulid species.

We now added results (from line 286/274) from studies on the effects of ocean acidification on shell growth of living and recent brachiopods. These have shown that brachiopods are generally resistant to lower pH values, have physiological buffering capacities for calcification and create suitable conditions at the site of calcification over a range of pH conditions (Cross et al. 2016, ICES Journal of Marine Science and Cross et al. 2018, Glob. Chang. Biol.).

We also now point out (from line 294/282) that we have chosen to interrogate the record of Soaresirhynchia and other rhynchonellid brachiopods precisely because these taxa survived the T-OAE event so can be analyzed to provide a continuous record before, during and after the T-OAE. Soaresirhynchia bouchardi was an opportunistic brachiopod species that dominates late T-OAE records. This suggests a notable resilience to the kill mechanisms of the extinction, which might be imparted by the ability to buffer internal pH and also survive the low oxygen conditions that occurred co-incident with the T-OAE. This has been well-documented for other extinctions events, such as the end-Permian, where biotas already adapted to low oxygen, such as those dominated by the bivalve Claraia, preferentially survived the extinction. Together, we used this as supporting evidence that the stable $\delta^{11}\text{B}$ values shown by the brachiopods, which contradict our B isotope record of micrite and the model, suggest that these taxa from the T-OAE studies were able to physiologically buffer changes in ocean pH and are hence of limited use for the interrogation of pH changes in Earth history.

We further discuss, that oysters (from line 312/300), similar to brachiopods, construct their shells of low-magnesium calcite, using the extrapallial fluid for calcification with the ability of regulating the calcifying fluid and maintaining stable pH conditions. All experimental studies on the response of the B isotope composition of oysters to changing ocean pH conditions support that oyster exert moderate-to-strong control over the pH of their calcifying fluid, keeping the pH of the extrapallial fluid elevated relative to the seawater pH under acidified conditions. The regulation of calcification site pH poses a limitation for the use of boron isotopes to reconstruct seawater pH from oysters and hence no B isotope-ocean pH calibration exists.

2. The timing of the $\delta^{11}\text{B}$ excursion in the two sections: The exact timing of the $\delta^{11}\text{B}$ excursion differs between the Portuguese and Spanish sections. In the Portugal section, the lowest pH is observed when the carbon isotope values are most negative, while in Spain, the lowest pH occurs toward the end of the T-OAE. This difference is difficult to reconcile with simple changes in sedimentary rates or cryptic hiatuses, and further explanation is needed. The authors could explore

this timing discrepancy in more detail and consider whether other factors, such as basin restriction or local depositional environment, might account for these variations.

The exact timing of the $\delta^{11}\text{B}$ excursion indeed differs, and so far, we have no definite explanation (line 397/354). The B isotope pattern for Portugal suggests a shorter-lived acidification event with a recovery within the T-OAE compared to Spain. The fact that in both sections the $\delta^{11}\text{B}_{\text{micrite}}$ values start to decrease at the time of the negative CIE, but show a different timing in the recovery, indeed suggests a basin restriction or different local depositional environments, as suggested by the reviewer.

As stated in the discussion (from line 397/354), differences in bathymetry, salinity and water temperatures between the two sites cannot explain the divergent pattern. Yet, different weathering intensities with different buffering capacities of the two sites coupled with basin restrictions might offer possible mechanisms. We initially referred to the Supplement (now “Environmental controls on the $\delta^{11}\text{B}$ composition of micrite” - Weathering) for further information on continental weathering fluxes based on osmium and strontium isotope studies and, in particular on calcium isotope information for the Lusitanian Basin (Portugal) and potential ocean pH buffering. We may not have been clear enough, but on the other hand we did not address this hypothesis in the manuscript any further because weathering rates for the Iberian Basin in Spain are not known and therefore, we cannot test it. We are currently investigating proxies for weathering to test this hypothesis, but this work is ongoing.

We however added some additional considerations in the revised version to be more up front with our hypothesis (line 401/358). Both sections are located in different basins on opposite margins of the Iberian Massif, which could lead to regional differences in weathering intensity. Increased continental weathering during the T-OAE with locally varying rates has already been proposed, with a fivefold increase observed for the Lusitanian Basin (Peniche) in Portugal based on calcium isotopes. The calcium and carbon isotope excursions show an obvious temporal synchronicity in the T-OAE, which also is reflected in our boron isotope excursion potentially indicating pH regulating in the ocean. While weathering could explain the divergent B isotope excursion, further investigations are needed because corresponding weathering rates are not known for the Iberian Basin in Spain. Indeed, there may be also differences in sedimentation rate, and cryptic hiatuses between the two localities, and the age models are not highly resolved (line 371/328). Nevertheless, both sets of data show a relationship between $\delta^{11}\text{B}$ values and the T-OAE CIE (line 373/331).

Additionally, there is a concern regarding the potential diagenetic overprint of the $\delta^{18}\text{O}$ data from micrite in Portugal, which may affect the interpretation of the $\delta^{11}\text{B}$ signal. The authors should consider how susceptible the $\delta^{11}\text{B}$ signal is to diagenetic alteration, particularly since the micrite samples are also subject to such processes. A discussion on the extent of diagenetic overprint and its potential impact on the $\delta^{11}\text{B}$ data would strengthen the manuscript.

While we have mentioned that our $\delta^{18}\text{O}_{\text{micrite}}$ values in both sections are lower than the comparable $\delta^{18}\text{O}_{\text{fossil}}$ values (line 170/161), we have in fact not discussed the potential diagenetic implications and how diagenesis may affect and alter the original chemistry of the micrite in the manuscript. In the revised version, we now point out from line 167/158 that the $\delta^{13}\text{C}$ values of the micrite argues for a good preservation, that $\delta^{18}\text{O}$ values indicate no significant deep burial alteration during lithification (line 173/163) and added (line 128/120 and from line 356/314) that diagenetic alteration of the B isotope composition can be ruled out. In the method section under “Potential influence of contamination and diagenesis on micrite” we describe the macroscopically (in the field and the

laboratory), microscopically (scanning electron microscope) and geochemically (carbon and oxygen isotopes, trace element) screening results of our used micrite and our assessment of post-depositional alteration. Especially meteoric diagenesis and recrystallization is assumed to decrease the isotopic composition of oxygen, boron and carbon isotopes. Our micrite $\delta^{13}\text{C}$ values overlap with published carbon isotope values for the Spanish and Portuguese sections from well-preserved brachiopods and bivalves and show the distinct carbon isotope trend found for the Pliensbachian-Toarcian time interval, arguing for a good micrite preservation. In contrast, the micrite $\delta^{18}\text{O}$ values are on average 1 ‰ lighter than the $\delta^{18}\text{O}$ from the fossils, pointing to a diagenetic overprint. However, $\delta^{18}\text{O}$ micrite values between -4.1 and -1.8 ‰ for Spain and -4.3 to -1.8 ‰ for Portugal, indicate no significant deep burial alteration during lithification and are consistent with published whole rock data from the Iberian Basin and the Lusitanian Basin.

We further discuss under “Potential influence of contamination and diagenesis on micrite” (from line 779/624), that the boron isotope pattern is both facies- and fabric-independent and similar in two comparable but independent transects from opposite margins around the Iberian Massif, indicating the preservation of a primary seawater isotope signature as e.g., late diagenetic features are expected to have high lateral variability and would disrupt the uniform and consistent trends seen in both sections. If we assume that the boron isotope excursion is a diagenetic feature, then we would have to consider a diagenetic front that affects only about 7 meters of the transect, roughly coincides with the T-OAE, cuts across different facies, and all this in marine carbonate sequences deposited in the NW Tethys in two different basins and at different water depths. Although such a scenario is not impossible, it is highly unlikely and we see no evidence for it.

Given these points, I recommend that the manuscript undergo major revisions to address these concerns before further consideration. I believe that with additional clarification and discussion of these issues, the manuscript could make an important contribution to the understanding of ocean pH and acidification during the T-OAE.

Below are some detailed comments and suggestions:

Line 32: delete “be”. Done

Line 48: change “negative large-scale” to “large-magnitude negative”. Done

Line 51: change “rising limb” to “recovery”. Done

Line 52-54: what about CO₂ removal by enhanced organic carbon burial? Please see comment below for Lines 206-208 and new Figure 9 in the supplement.

Line 110: removal “the same” Done

Lines 138-140: I don’t see any decrease in $\delta^{18}\text{O}$ values from micrite for the Portugal record, but rather a slight increase. What does this mean in terms of diagenetic overprint on $\delta^{18}\text{O}$ as well as $\delta^{11}\text{B}$ values? We describe the pattern of the brachiopods and bivalves and not of the micrite in this paragraph. The description of the micrites follows below. For the potential diagenetic overprint, please see our explanation above.

Lines 144-147: due to the data gap in most of the T-OAE, an alternative way to interpret the $\delta^{11}\text{B}$ data of bivalves (Spain) is that the data in general fits with the values and trends generated from micrite. That might be the case considering that the bivalves (Spain) show $\delta^{11}\text{B}$ values with an average of $+12.3 \pm 0.7$ ‰ and the micrites of $+13.0 \pm 1.5$ ‰, but there is only one data point close to 0 meters that roughly overlaps.

Lines 150-153: following the above comment, the $\delta^{18}\text{O}$ micrite values are lighter, but also different trends than the $\delta^{18}\text{O}$ fossil values. Could this indicate diagenetic overprint on the sample? As we explain above, the micrite is lighter in its oxygen isotopic composition than the fossils as one would expect during post-depositional alteration during the conversion from unlithified lime mud into micrite. But the data and the screening indicate no significant deep burial alteration during lithification and meteoric diagenesis.

Line 160: for “subsequent”, do you mean “stratigraphically above”? Yes, and we changed it.

Line 165: change “middle and early Toarcian” to “early and middle Toarcian”. What does the word “respectively” refer to? We deleted respectively and changed middle and early.

Line 166: change “before” to “prior to”. Done

Lines 206-208: what is the role of organic carbon burial in bringing the system back here? It has been suggested to be an important carbon sink for this event, especially with the widespread organic-rich black shale deposition at this time.

Organic carbon burial is a CO_2 sink, because greenhouse CO_2 is reduced then buried in rock (please see equation (7), Supplementary Model) and a net oxygen source, because reduced organic carbon, fixation of which released oxygen in 1:1 stoichiometry, escapes respiratory oxidation (original COPSE reloaded paper, differential equation for oxygen, table 1, page 5:

$$\frac{d\text{O}}{dt} = \text{locb} + \text{mocb} - \text{oxidw} - \text{ocdeg} + 2 \cdot \text{mpsb} - 2 \cdot \text{pyrw} - 2 \cdot \text{pyrdeg}$$

where all flux abbreviations are as described in the supplement, “mocb” denoting marine organic carbon burial).

The greenhouse input drives a nutrient input, therefore a production and mocb increase, as described. Additionally, please also see new supplementary figure 9, which corresponds to the results given in the main text (i.e., isotopically neutral CO_2 , -60 methane).

The format of this figure (supplementary figure 9) is identical to the other supplementary figures, with the addition of two extra panels, bottom row, corresponding to the fluxes in the organic and carbonate carbon derivatives, each normalized to their present value. Note in particular panel I, bottom left, blue line – illustrating a significant increase in marine organic carbon burial across the modelled OAE, conforming with the black shale deposition and equivalent data, as the reviewer suggests.

Note that aerobic respiratory breakdown of marine organic carbon is represented in the COPSE model, meaning that under anoxic conditions marine organic carbon burial will increase because less production gets respired in the water column, meaning more ends up getting buried – until mocb’s above impact on the global oxygen reservoir reduces anoxia in a net negative feedback (which partly controls the length of the OAE). We have not laboured this point in the main text, because this basic feedback sequence is well-appreciated in literature pertaining to models of this sort, and not of direct relevance to the ocean acidification signature issue.

Line 277: delete “is”. Exchanged “is” with “occurs”

Line 278: how does differences in sedimentation rate influence the relative timing of pH change to the T-OAE? This was to find potential explanations for the offsets in the B isotope pattern and sedimentation could have an influence on the correlation regarding our resolution. As mentioned above, we expanded on this at the end of this part of the discussion from line 399/356.

Lines 297-299: The $\delta^{11}\text{B}$ data shows a recovery to pre-Toarcian conditions during the Serpentinum zone, instead of at the end. *We changed it.*

Lines 331-332: depending on how you look at the data, it is also possible to say that the minimum pH coincides with the abrupt increase in shell size. *We changed the wording (line 433/390). The discrepancy between still low pH and rapid recovery in shell size, as here referred to by the reviewer, was already pointed out in the next paragraph of the main text (from line 440/397).*

Supplementary materials:

Age model: why the Geological Time Scale 2016 is used, instead of the more recent ones. *We started the project and the modelling already some time ago and hence with the Geological Time Scale 2016. We realize that there is a more recent one, but an age model is in a state of flux and the minor changes between the 2016 and 2020 one will not change our model output or our message.*

Line 217: should it be “cannot” or “can” account for the offset? *It is indeed “cannot”.*

Reviewer #3 (Remarks to the Author):

Review of Kasemann et al.

The manuscript by Kasemann et al. makes the case that the B-isotope composition of micrite provides a reliable and robust indicator of ocean pH changes in deep time, and that the data obtained across the T-OAE reveal a signal of acidification coincident with carbon release and warming. The work is very well-written and the illustrations are good. The manuscript is well-structured and well-referenced. Overall, the manuscript provides important and novel data and I recommend publication after minor revision.

Main comments

I am not an expert in B-isotope systematics, so I cannot find fault with the technical details of the work. I have only minor comments on the modeling approach and some of the text relating to the way the T-OAE is presented.

One issue, which is admittedly minor in the context of the work, is that I do think we need to move away from lumping the Karoo and Ferrar LIPs into a single entity (e.g., lines 41, 285 etc.). These LIPs were distinct in onset timing and genesis, and recent work has indicated that it was Ferrar that was coeval with the T-OAE, and Karoo started earlier and likely initiated climate changes at the PI-To. The literature citing Karoo as the event coincident with the T-OAE did so because of the incorrect assumption that the T-OAE was older than we actually now know it was (see Al-Suwaidi et al., 2022; Kemp et al., 2024).

We changed it in the text and also now referenced Kemp et al., 2024

I'm not a huge fan of the approach the authors use for source modeling. In all experiments, a biogenic CH_4 component is assumed (-60%) along with a volcanic CO_2 component. I think it would have been more sensible to avoid assuming a mix of 2 specific C sources and instead derive a single average value for the likely isotopic composition of the total C. From that, the likely contribution of

different sources (biogenic, volcanic etc.) could be discussed later. At present, the results and figures show how a specific balance of carbon from hydrate CH₄ (−60‰) and volcanic CO₂ could fit the data. But couldn't the same results be achieved if the carbon was a different mix of volcanic CO₂ and thermogenic CO₂/CH₄ (i.e. from sill intrusion into organic-rich rocks, ~−30‰)?

Part of the reason I ask is because the potentially important role of thermogenic C released during sill intrusion is effectively ignored (see for example: McElwain et al., 2005; Svensen et al., 2007, 2012; Heimdal et al., 2021; Kemp et al., 2024). Average C source compositions of −11‰ and −15‰ are implied by cGENIE modelling for the PETM and Permian-Triassic, and these values are perhaps most likely to have been achieved by a mix of volcanic and thermogenic sources – without the need for any/much biogenic CH₄ (Gutjahr et al., 2017; Cui et al., 2021). This is not to say methane was not involved during the T-OAE (the astronomical pacing of the CIE supports it), but it just seems like an oversight to ignore a thermogenic contribution. Maybe too late now to change anything since the models have been run, but my suggestion is that the authors maybe note the assumptions being made about the mix of two specific sources, and add a line or two to note that the carbon could have been a mix of volcanic and thermogenic (if the authors agree that this is reasonable).

The purpose of the greenhouse inputs in this model is to simultaneously produce the large scale positive $\delta^{13}\text{C}$ excursion (via the weathering/productivity associated model feedbacks discussed in the main text Modelling approach from line 197 tracked changes and from line 184 in the cleaned version, which is interspersed with the brief negative $\delta^{13}\text{C}$ excursion. The greenhouse inputs selected for the main model run (isotopically neutral CO₂, clathrate methane with composition of -60 ‰) are not the only way to reproduce the data, e.g., see supplementary figures 4-6. However, regardless of the specific choice of greenhouse forcings, the general conclusion that a substantial ocean acidification event occurs and is detectable in the $\delta^{11}\text{B}$ data, is robust.

We include additional supplemental figures 7 and 8, respectively showing zero CH₄ input, and CH₄ input with a composition of -30 ‰, in both cases in conjunction with isotopically neutral CO₂. In the former case the negative limb of the $\delta^{13}\text{C}$ data cannot be reproduced. This illustrates why at least some methane must be included – in order to get a sudden negative excursion, from a baseline state of positive excursion caused by the CO₂ input. In the latter case (supplementary figure S8, methane at -30 ‰), the baseline methane flux must be approximately doubled in order to reproduce the lower point of the data, which results in a stronger OAE overall.

In summary then, the modelling approach used here suggests that some methane must be involved, but that various combinations of isotopic composition/quantity can fit the data. Nevertheless, all result in the acidification event discussed, which is the main focus here.

Line by line comments

Line 43. “as high as”. Both Ruebsam et al. (2020) and McElwain et al. (2005) have data to suggest it was higher than 900 ppm. *We changed this in line 45/43.*

Line 51. Might be good to consider alternative wording here. “rising limb...” may not mean much to the reader, and maybe better to define it in terms of a stabilization of values or the end of a positive shift etc. *We changed it to “recovery” as suggested by reviewer 2.*

Line 53. Not that ‘recent’ now. Even more recently, a CIE duration of ~300 kyr (and almost certainly <407 Kyr) has been constrained by CA-ID-TIMS geochronology (Kemp et al., 2024). Too late now for

the modelling, but would be interesting to know whether this dramatically effects things. Maybe a line could be added to briefly note the implications of a faster release of carbon than the manuscript currently assumes? *Our new figure 8 in the supplement us is covering this point. We basically get a stronger anoxia event (i.e., greater % ocean waters going anoxic).*

Line 110: What makes them 'well-preserved' successions?. *We deleted "well-preserved" and added that "These show no evidence of widespread recrystallisation, deep-burial overprint, dolomitization, or meteoric diagenetic influence" (line 127/119).*

Line 131: "We place all data into the same biostratigraphic...". Repetition of something already clear in the last section. *We deleted the sentence.*

Line 145, 161, 231: 'data gaps'. Because of the lack of suitable material, lack of any material, lack of sampling, or lack of good outcrop? I know Methods is cited here but a single line in the main text to explain why we have these gaps could be helpful. *It is indeed lack of suitable material that did not pass our screening protocol and we added this in the main text (line 164/154, 184/173).*

Al-Suwaidi, A.H. et al., 2022. New age constraints on the Lower Jurassic Pliensbachian–Toarcian Boundary at Chacay Melehue (Neuquén Basin, Argentina). *Scientific Reports*, 12, 4975.

Cui, Y. et al., 2021. Massive and rapid predominantly volcanic CO₂ emission during the end-Permian mass extinction. *Proc. Natl. Acad. Sci. U.S.A.*, 118, e2014701118.

Gutjahr, M. et al., 2017. Very large release of mostly volcanic carbon during the Palaeocene–Eocene Thermal Maximum. *Nature*, 548, 573–577.

Heimdal, T.H., 2021. Assessing the importance of thermogenic degassing from the Karoo Large Igneous Province (LIP) in driving Toarcian carbon cycle perturbations. *Nature Communications*, 12, 6221.

Kemp, D.B. et al., 2024. The timing and duration of large-scale carbon release in the Early Jurassic. *Geology*, doi:10.1130/G52457.1.

McElwain, J.C. et al., 2005. Changes in carbon dioxide during an oceanic anoxic event linked to intrusion into Gondwana coals. *Nature*, 435, 479–495.

Ruebsam, W. et al., 2020. $\delta^{13}\text{C}$ of terrestrial vegetation records Toarcian CO₂ and climate gradients. *Scientific Reports*, 10, 117.

Svensen, H. et al., 2012. Rapid magma emplacement in the Karoo Large Igneous Province: Earth and Planetary Science Letters. 325–326, 1–9.

Svensen, H. et al., 2007. Hydrothermal venting of greenhouse gases triggering Early Jurassic global warming. *Earth Planetary Science Letters*, 256, 554–566.

RE: “Ocean acidification at the Toarcian Anoxic Event captured by boron isotopes in the lime mud record” by Kasemann et al.

We would like to thank the reviewers for their positive assessments and we welcome the additional point raised by Reviewer #2.

To facilitate the evaluation of the changes made, we have copied the comment of Reviewer #2 in blue/non-italics and provided the response in black/italics. All line numbers cited in our responses refer to the revised version of the manuscript.

Reviewer #2 (Remarks to the Author):

I appreciate the authors' efforts in thoroughly addressing the reviewers' previous comments. The manuscript has significantly improved as a result. The authors have clarified that the boron isotope data from micrite and carbonate macrofossils are used to support the modelled results, and they now provide a more detailed discussion of vital effects in brachiopod and bivalve shells—specifically explaining why these organisms may not record pH changes as expected.

However, it would strengthen the argument to further elaborate on why micrite is considered a reliable archive of pH. As noted in the introduction, micrite is thought to originate primarily from algae, microbes, or more likely from calcareous nannoplankton (e.g., *Schizosphaerella*, coccoliths, dinoflagellates), and may also include admixtures of peri-platform ooze. It would be helpful if the authors could provide supporting evidence or references demonstrating that these sources are capable of recording seawater pH without significant influence from biological vital effects.

I recommend publication pending minor revisions to address this point.

One aspect that we emphasize in our manuscript is that we do not consider micrite as a reliable archive of pH to obtain estimates of absolute ocean pH values and absolute changes in ocean pH. Instead, we argue that we can use the B isotope composition of micrite as an archive to track changes in pH. The manuscript states: “...micrite... have been demonstrated to offer a reliable archive in deep time for measuring B isotope values to track trends in ocean pH conditions” (line 94), “...micrite B isotope values are inferred to more faithfully track ocean pH changes” (line 414), and “we can nonetheless show here that B isotope values derived from micrite can captures trends in ocean pH changes” (from line 419). To emphasize this further, we have added the word ‘relative’ to the sentence in line 414.

*That we want to track the relative changes in ocean pH is addressed in the introduction just before the statement (line 93) that we assume that the micrite originates primarily from algae, microbes or, more likely, calcareous nannoplankton (e.g., *Schizosphaerella*, coccoliths, dinoflagellates) and may also contain admixtures of peri-platform ooze (from line 96). It is not known whether *Schizosphaerella* or dinoflagellates are able to record the pH value of seawater with or without being significantly influenced by vital effects. However, studies on the B isotope composition of cultured and wild-grown coralline algae found a significant relationship between the $\delta^{11}\text{B}$ values and the pH of the seawater, but also observed an up-regulation of the calcifying fluid pH (Donald et al. 2017, Piazza et al. 2023) that is potentially species-specific (Cornwall et al. 2017). A study of the B-isotope composition of coccolithophores by Liu et al. (2021) revealed different species-specific behaviours in the regulation of the pH of the calcifying fluid, where in addition to homeostasis, a relationship between changes in seawater pH and $\delta^{11}\text{B}$ values in the coccolith calcite was observed. In addition to Paris et al. (2010),*

we also successfully analyzed microbial and abiotic micrites in the Neoproterozoic (Kasemann et al. 2010, Ohnemüller et al. 2015) and the PTB (Clarkson et al. 2015) and found a significant relationship between the $\delta^{11}\text{B}$ values and the pH of the seawater that in case of the PTB is considered to record seawater $\delta^{11}\text{B}$ values in the absence of vital effects (Clarkson et al. 2015). As such we believe that micrite is capable of recording seawater pH when properly screened (method section from line 446), but there is indeed the possibility of an influence from biological vital effects. We have included this information in the Supplementary Discussion in the section “Environmental controls on the $\delta^{11}\text{B}$ composition of micrite - source material” for clarification.

In conclusion, the pattern of the modelled ocean pH was found to be broadly consistent with the micrite B isotope patterns, supporting our hypotheses that ocean pH is the major driver for the observed $\delta^{11}\text{B}$ changes in the micrite. Considering the potential heterogeneous mixture of the micrite of material that can record changes in ocean pH, we cannot exclude vital effects on the B isotope composition of the calcite material and hence concluded that “Unfortunately, there is no useful $\delta^{11}\text{B}$ ocean pH calibration for any of these sample types “. To clarify this, we changed the sentence and referred to the Supplementary Discussion for further information “While relationships between the $\delta^{11}\text{B}$ value of calcite and seawater pH have been observed for some of the assumed source materials in the micrite (Supplementary Discussion), there is no useful $\delta^{11}\text{B}$ ocean pH calibration for this mixture of sample types, and we can therefore only track trends in ocean pH conditions" (from line 104).

As pointed out in our first rebut to Reviewer #1, we are quite aware of the problematic nature of using micrite as an archive for $\delta^{11}\text{B}$ -ocean pH and this is why we, in this study, took a different approach to test ocean acidification and the ability of the micrite to track changes in ocean pH. Normally, one produces a B isotope record from carbonate skeletal material (e.g., forams, corals, micrite) and calculates an ocean pH record. We have also done this for the Neoproterozoic and for the PTB using micrite. In this study, we started by reconstructing a record of ocean pH, carbon cycle disturbance and rapidly changing environmental conditions using the COPSE model and previously published data e.g., carbon isotopes. We have tried to make this more explicit in the first review round of the revised text (from line 106).

We also pointed out to Reviewer #1 in the first round, that algae, microbes or most likely calcareous nannoplankton that may have produced the majority of the micrite analyzed is to some extent of course a quite heterogeneous collection of material, but on a much smaller spatial scale it seems to homogenize the collected environmental signal. This may seem contradictory at first, but in principle it is exactly what we do when we use foraminifera or corals as archives. We know from in-situ techniques such as laser and ion probe that the B isotope composition in a foraminifera test (from chamber to chamber) and in a coral (sometimes in micrometer scale) is usually highly variable. But once we have a representative sample that we analyze wet-chemically with TIMS or the MC-ICP-MS, we homogenize the values and can use the skeletal carbonate material as an archive for B isotope-ocean pH calibrations. The same also applies to other isotope systems such as carbon and oxygen isotopes.

Minor suggestions (line numbers from the tracked changes version)

Line 22: it would be good to spell out Ma in first use. We changed this in the abstract.

Line 33: what does interrogated taxa mean? It also does not read well to have interrogated and interrogation in the same sentence. We agree and changed it to “the investigated taxa”.

Line 50: change “found” to “observed” We changed it.

Lines 50-51: change to “observed in fossil wood, diverse marine bulk organic and inorganic substrates, and carbonate macrofossils” *We changed the sentence.*

Lines 131: add “soly” before “using the B isotope records” *We added “solely” to the sentence.*

Line 184: delete “did not” *We deleted it, thanks for pointing out.*

Line 301: change “extinctions events” to “extinction events” *We changed it.*

References:

Clarkson, M. O., Kasemann, S. A., Wood, R. A., Lenton, T. M., Daines, S. J., Richoz, S., OhnemueLLer, F., Meixner, A., Poulton, S. W. & Tipper, E. T. Ocean acidification and the Permo-Triassic mass extinction. *Science* **348**, 229-232 (2015).

Cornwall, C. E., Comeau, S., McCulloch, M.T. Coralline algae elevate pH at the site of calcification under ocean acidification. *Glob Change Biol.* **23**, 4245-4256 (2017).

Donald, H.K, Ries, J.B., Stewart, J.A., Fowell, S.E, Foster, G.L. Boron isotope sensitivity to seawater pH change in a species of *Neogoniolithon* coralline red alga. *Geochimica et Cosmochimica Acta.* **217**, 240-253 (2017)

Kasemann, S. A., Prave, A. R., Fallick, A. E., Hawkesworth, C. J. & Hoffmann, K.-H. Neoproterozoic ice ages, boron isotopes, and ocean acidification: Implications for a snowball Earth. *Geology* **38**, 775-778 (2010).

OhnemueLLer, F., Prave, A. R., Fallick, A. E. & Kasemann, S. A. Ocean acidification in the aftermath of the Marinoan glaciation. *Geology* **42**, 1103-1106 (2014).

Paris, G., Bartolini, A., DonnadiEU, Y., Beaumont, V. & Gaillardet, J. Investigating boron isotopes in a middle Jurassic micritic sequence: Primary vs. diagenetic signal. *Chem. Geol.* **275**, 117-126 (2010).

Piazza, G., Paredes, E., Bracchi, V. A., Pena, L. D., Hall-Spencer, J. M., Ferrara, C., Cacho, I., Basso, D. Multi-specific calibration of the B isotope proxy in calcareous red algae for pH reconstruction, EGU General Assembly 2023, Vienna, Austria, 24–28 Apr 2023, EGU23-12561, <https://doi.org/10.5194/egusphere-egu23-12561>, 2023.

Liu, Y.-W., Rokitta, S.D., Rost, B., Eagle, R.A. Constraints on coccolithophores under ocean acidification obtained from boron and carbon geochemical approaches. *Geochimica et Cosmochimica Acta.* **315**, 317-332 (2021).